# Genetic population structure of *Haemophilus influenzae* at local and global scales

Neil MacAlasdair [1,2,11], Anna K. Pöntinen [1,11], Clare Ling[3,4,11], Sudaraka Mallawaarachchi [1,5,6], Janjira Thaipadungpanit [7], Francois H. Nosten [4,8], Claudia Turner[3,4], Stephen D. Bentley [2], Nicholas J. Croucher [9], Paul Turner [3,4,12] ✉ & Jukka Corander [1,2,10,12] ✉

*Haemophilus influenzae* is an opportunistic bacterial pathogen that causes both non-invasive and invasive disease in humans. Although the *H. influenzae* type b vaccine can reduce invasive disease, it is not effective against non-b serotypes or unencapsulated non-typeable *H. influenzae* (NTHi). The genetic population structure of *H. influenzae*, especially NTHi, which is typically prevalent in lower- and middle-income countries, is unclear. Here we whole-genome sequenced 4,474 isolates of *H. influenzae* from an unvaccinated paediatric carriage and pneumonia cohort from the Maela camp for displaced persons in northwestern Thailand. Despite no *H. influenzae* type b immunization, serotype b was uncommon, whereas 92.4% of the isolates were NTHi. Most multidrug-resistant lineages were NTHi, and there were no lineages enriched among disease samples. Incorporating 5,976 published genomes revealed a highly admixed population structure, low core genome nucleotide diversity and evidence of pervasive negative selection. Our findings expand our understanding of this major pathogen in lower- and middle-income countries and at a global scale.

The nasopharynx is the natural habitat of the bacterium *Haemophilus influenzae*, where it exists in asymptomatic carriage. However, it frequently translocates to other body sites such as the inner ears, lungs and sinuses, causing a range of disease manifestations[1]. The most common of these is acute otitis media (AOM), which is one of the leading causes of antibiotic prescriptions in children. Global estimates suggest over 700 million AOM cases per annum caused in total by any bacterial pathogen, and a substantial fraction of these lead to further complications and sequelae, particularly in low- and middle-income countries (LMICs)[2]. After licensing of the polysaccharide-protein conjugate *H. influenzae* type b (Hib) vaccine in the late 1980s, its adoption into national vaccination programmes worldwide has led to a notable reduction of Hib colonization and its associated invasive disease manifestations, such as meningitis and pneumonia. However, the vaccine

does not protect against colonization by other serotypes or unencapsulated non-typeable *H. influenzae* (NTHi). Therefore, *H. influenzae* remains a major cause of AOM, sinusitis, conjunctivitis and pneumonia and consequently is an important public health burden globally. A particular concern has arisen from the widespread antibiotic resistance observed in some strains of NTHi and particularly the possibility of multidrug resistance (MDR), as widespread β-lactamase resistance has led to difficulties in treating recurrent infections, particularly in children where alternate classes of antibiotics may not be approved[3].

The evidence for NTHi as an important cause of paediatric community-acquired pneumonia (CAP) has been summarized in comprehensive reviews[4,5]. The determination of aetiology in paediatric CAP remains a challenge, with a minority of cases being bacteraemic. However, specimens obtained via bronchoscopy revealed NTHi to be

**Fig. 1 | Overview of the Maela cohort study design and the global collection of published genomes. a**, The geographical location of the study site. **b**, The cohort design and sample processing for 999 mother–infant pairs recruited to the study. The nasopharyngeal (NP) swabs were taken at monthly health checks (indicated by black asterisks) and at any timepoint when the infant presented symptoms of clinical pneumonia (indicated by red asterisks). Bottom: the sample numbers at each step of the study. **c**, The global map coloured by the number of isolates per country of origin in the systematic public collection of global *H. influenzae* isolates.

the dominant bacterial pathogen in 250 Belgian children with recurrent or non-resolving CAP[6]. The nasopharyngeal colonization by non-Hib/NTHi, especially at higher densities, has also been shown to be associated with paediatric CAP in LMICs[7,8].

Whole-genome sequencing (WGS) studies of *H. influenzae* have been mainly conducted from smaller-scale collections of disease cases[9,10] but rarely from large-scale collections of both carriage and disease isolates of the same population. Furthermore, few studies have been conducted in LMICs, where nasopharyngeal pathogen colonization rates and the burden of CAP are generally much higher than in high-income countries. As a consequence, the genetic population structure and evolutionary dynamics of the species remain poorly understood in LMIC settings and at a global scale[11].

This motivated us to conduct a longitudinal paediatric cohort study of both healthy colonization and pneumonia among a large birth cohort in a population located in northwestern Thailand, the Maela camp for displaced persons. The densely populated camp is located on the Thailand–Myanmar border and provided a unique opportunity to systematically sample both carriage and disease cases in a pre-Hib vaccine population. Here, we detail results from the WGS of isolates from our cohort and also of further analyses performed on the Maela data combined with all publicly available high-quality *H. influenzae* genomes with sufficient metadata. This combined collection of 9,849 genomes allowed us to conduct genomic analyses of the species at a global and species-wide scale, providing novel insight into how its high levels of recombination shape its global population structure.

## Results

### Serotype distribution across the Maela paediatric population

The infants in the Maela cohort carrying *H. influenzae* (Fig. 1a,b) were predominantly colonized by NTHi, despite lacking immunization against Hib (Table 1). Out of 3,970 isolates that passed the final quality control (QC) filters, 613 (15.4%) were from cases of pneumonia. Of the 3,210 host-deduplicated isolates, 524 were collected from pneumonia cases (16.3%). The counts and estimated frequencies of the six different serotypes and non-typeable (NT) (unencapsulated) isolates are listed both with and without host deduplication in Table 1. Notably, NT isolates made up 91.7% of all isolates, and serotype b isolates are the second most prevalent, making up 5.7% of the population. The remaining five serotypes account for less than 1% of the population each. The serotypable isolates generally form monophyletic lineages on the tree (Fig. 2). Two isolates, one NT and one serogroup b by agglutination, gave only partial in silico capsule typing results due to us being unable to identify the entire capsule locus in the data. The in silico capsule typing was generally congruent (overall congruence 95.3%) with the agglutination-based phenotypic serotyping (except for serotypes d and e) and was corrected by the latter in those 28 cases where the serological typing indicated a serotype for a NT in silico type. These included serotype a (*n* = 1), b (*n* = 19), d (*n* = 3) and e (*n* = 5). Our results are reasonably well in line with earlier comparisons between agglutination-based and in silico typing (98–100% congruence)[12,13]; however, the higher level of discrepancy observed here could be due to the much larger and more diverse set of genomes considered. The population frequencies of

**Table 1 | Proportions and counts of the number of isolates of each serotype and non-typable isolates in the Maela collection and within pneumonia and non-pneumonia cases**

| Serotype | Maela collection (n=3970) | | | Host-deduplicated isolates (n=3210) | | |
|---|---|---|---|---|---|---|
| | Number of isolates (percentage of total, 3,970) | Pneumonia cases (percentage of total, 613) | Non-pneumonia cases (percentage of total, 3,357) | Number of isolates (percentage of total, 3,210) | Pneumonia cases (percentage of total, 523) | Non-pneumonia cases (percentage of total, 2,687) |
| a | 15 (0.4%) | 0 | 15 (0.4%) | 11 (0.34%) | 0 | 11 (0.4%) |
| b | 208 (5.2%) | 19 (3.1%) | 189 (5.6%) | 154 (4.8%) | 15 (2.9%) | 139 (5.2%) |
| c | 6 (0.2%) | 2 (0.3%) | 4 (0.1%) | 6 (0.19%) | 2 (0.4%) | 4 (0.1%) |
| d | 5 (0.1%) | 2 (0.3%) | 3 (0.1%) | 5 (0.16%) | 2 (0.4%) | 3 (0.1%) |
| e | 31 (0.8%) | 5 (0.8%) | 26 (0.8%) | 26 (0.81%) | 4 (0.8%) | 22 (0.8%) |
| f | 36 (0.9%) | 2 (0.3%) | 34 (1.0%) | 31 (0.97%) | 2 (0.4%) | 29 (1.1%) |
| Non-typeable | 3,669 (92.4%) | 583 (95.1%) | 3,086 (91.9%) | 2,977 (92.7%) | 498 (95.2%) | 2,479 (92.3%) |

The serotypes were determined with both agglutination and in silico using sequence data (Methods). Owing to the longitudinal nature of the sampling, host deduplication was used to remove isolates which were collected from the same host on consecutive sampling times and were likely to be clonally related (Methods).

serotypes were highly similar between pneumonia and non-pneumonia cases. The distribution of serotypes in pneumonia/non-pneumonia cases is as follows, in pneumonia: 583 NT (95.1%), 19 serotype b (3.1%), 5 serotype e (0.8%) and 2 (0.3%) each of serotypes c, d and f; and in non-pneumonia: 3,086 NT (91.9%), 189 serotype b (5.6%), 34 serotype f (1.0%), 26 serotype e (0.8%), 15 serotype a (0.4%), 4 serotype c (0.1%) and 3 serotype d (0.10%) (Table 1).

### Genetic population structure of *H. influenzae* in Maela

Core genome multilocus sequence type (cgMLST), has previously been used to study *H. influenzae*[11], but when applied to the entire global dataset (Fig. 3), neither the typing nor clustering by allelic profiles was able to provide meaningful insight into the population structure of *H. influenzae* in the Maela cohort owing to too-high diversity in the allelic profiles. Of note, the number of cgMLST allelic profiles and the overall nucleotide diversity are not always strongly correlated, as many very low-frequency mutations can generate a large number of allelic profiles, although nucleotide diversity across the pangenome remains low. The PopPUNK clustering method, which combines information from core and accessory genomic variation, largely identified monophyletic clusters among Maela isolates (Fig. 2). However, the largest cluster contained 349 isolates, with only 13 PopPUNK clusters consisting of at least 100 isolates (out of a total of 122 clusters) and 20 clusters consisting of 50 or more isolates. A large proportion of clusters (50%) contained ten or fewer isolates. The serotypable isolates were found in ten clusters in total (serotypes a, b, c and f each in two clusters, with serotypes d and e each in one cluster). NTHi were observed as the dominating type in both non-pneumonia and pneumonia timepoint samples (Fig. 2), and no particular genetic lineage of NTHi was overrepresented in pneumonia samples (Fisher's exact test *P* value 0.091, Methods).

### Distribution of AMR determinants

Antimicrobial resistance (AMR) determinants were frequently identified across the phylogeny and strongly associated with MDR lineages (Methods; Fig. 2). Only one of the MDR lineages was clearly associated with serotype b (Fig. 2), and the remainder were NTHi. A total of 41 PopPUNK (Methods) clusters contained at least one MDR isolate, indicating the repeated acquisition of AMR determinants across the population. In the Maela host-deduplicated dataset, most of the MDR (resistance against at least four out of nine antibiotic classes; Methods) isolates (507/3,210, 15.8%) were NT (77.3%, 392/507), followed by serotype b (22.3%, 113/507) and two serotype e isolates (0.4%). Hence, serotype b was clearly overrepresented among the more resistant isolates (overall frequency 4.8%; Table 1), whereas there were less NT (overall frequency 92.7%; Table 1) and a high proportion (73.4%) of serotype b isolates are MDR (113/154) compared with NT (392/2,977, 13.2%) or serotype e (2/26, 7.7%).

Within pneumonia cases (523/3,210), 17.0% (89/523) of isolates were MDR, of which 76 were NT, 12 were serotype b and 1 was serotype e, among 498 NT (95.2%), 15 serotype b (2.9%), 4 serotype e (0.8%), 2 serotype c (0.4%), 2 serotype d (0.4%) and 2 serotype f (0.4%) isolates. Within non-pneumonia cases (2,687/3,210), 15.6% (418/2,687) were MDR, of which 316 were NT, 101 were serotype b and 1 was serotype e, among 2,479 NT (92.3%), 139 serotype b (5.2%), 29 serotype f (1.1%), 22 serotype e (0.8%), 11 serotype a (0.4%), 4 serotype c (0.2%) and 3 serotype d (0.1%) isolates. Therefore, the frequency of the MDR phenotype was highly similar between the two sample types.

### Quantification of homologous recombination

Because the acquisition of AMR determinants is probably aided by horizontal gene transfer in this naturally transforming species, we quantified the extent of homologous recombination. Mapping Illumina reads from isolates in the same PopPUNK cluster against long-read reference assemblies to produce whole-genome pseudoalignments to be used as inputs to single nucleotide polymorphism (SNP)-density based recombination analysis (Gubbins) was not a feasible approach to quantify recombination in the entire Maela cohort owing to the size of the dataset and the large number of PopPUNK clusters present.

Consequently, we leveraged the aligned pangenome genes for the 3,970 Maela isolates to perform per-gene recombination inference (Methods). Of the 7,015 genes in the inferred pangenome, at least one recombination event between PopPUNK lineages was identified in 2,672 genes (38%). On average, 193.36 recombination events were identified per gene (including recombination-free genes), and the frequency of recombination events was significantly correlated with the estimated nucleotide diversity per gene (Spearman's $r = 0.49$, $P < 7.15 \times 10^{-293}$). A substantial proportion of genes with no detected recombination (64.0%) also had zero nucleotide diversity (Extended Data Fig. 1a). Finally, we also quantified the rate of decay of linkage disequilibrium in the core genome and compared this with several other common bacterial pathogens analysed in ref. 14. This showed that the decoupling of SNPs as a function of base-pair distance happens fastest in *H. influenzae* (Extended Data Fig. 1b), and the rate is considerably elevated compared with other species known to routinely engage in homologous recombination, such as *Campylobacter jejuni* and *Enterococcus faecalis*. Taken together, these results suggest that the *H. influenzae* population within Maela is extremely recombinant, to the extent that it probably reduces the overall level of diversity within the population.

### Population genetic analyses of the global dataset

To understand how the genetic variation observed in the Maela cohort samples relates to internationally circulating *H. influenzae*, we combined the study data with a systematic collection of all publicly available

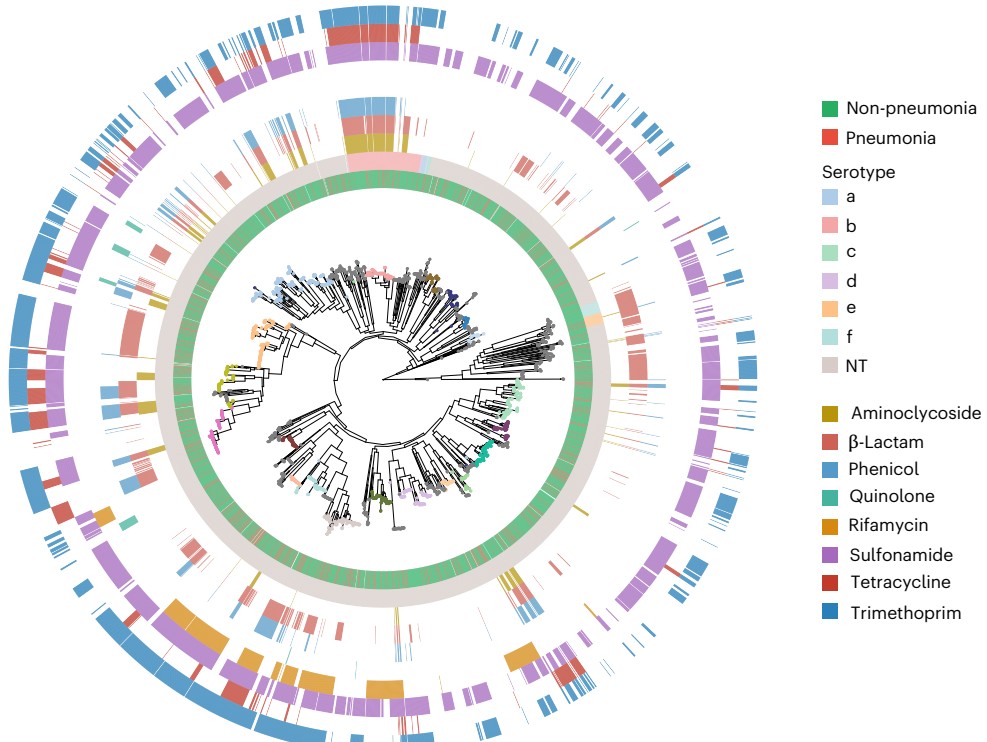

**Fig. 2 | Phylogeny of Maela *H. influenzae* genomes for 3,970 isolates.** The sample type is indicated by the innermost ring. The phylogeny was estimated using FastTree v.2.1.10 on the core-genome alignment mapped against the *H. influenzae* reference 86-028NP (NC_007146.2). The 20 largest PopPUNK clusters (>50 isolates) are indicated by coloured dots at the tips of the phylogeny, whereas smaller clusters are in grey. The in silico serotypes (second ring), inferred by using Hicap v.1.0.3, and AMR profiles (eight outer rings), screened with AMRFinderPlus v.4.0.3, are shown by colour as indicated in the legend. An interactive online phylogeny, with additional metadata including cgMLST and cgMLST cluster data, is available at ref. 84.

*H. influenzae* genome data with basic metadata available (country and year of collection), for a total dataset comprising 9,849 isolates (Methods; Fig. 1c). The PopPUNK clustering of the combined dataset identified 752 monophyletic lineages, with the largest composed of 483 isolates (>99% serotype a). There were 20 clusters with at least 100 isolates and 595 clusters with <10 isolates. Many larger clusters were paraphyletic according to the core genome phylogeny (Fig. 3), whereas the monophyletic lineages mostly corresponded to small or singleton clusters.

The core genome phylogeny of the global collection clearly demonstrates that the Maela isolates are extensively interspersed within the global population of the species, suggesting a rapid cross-border and intercontinental transmission. Furthermore, the isolates spanning the sampling window (1962–2023) are distributed across the phylogeny and do not form monophyletic lineages made up of temporally restricted isolates. Together, these patterns strongly suggest suggest that the history of migration within the global *H. influenzae* population is sufficiently frequent and extensive to erase any phylogeographical signal of the local clonal expansion of lineages.

Given the low level of nucleotide diversity identified in the recombination analysis of the Maela cohort, we further investigated the overall level of nucleotide diversity across the aligned pangenome of the entire global collection, which consisted of 18,265 genes. Although the nucleotide diversity in both core ($n = 1103$) and non-singleton accessory ($n = 8843$) genes have overlapping ranges (Fig. 4a), core genes are on average significantly less diverse than accessory genes (two-tailed Mann–Whitney $U$ test, $P = 1.205 \times 10^{-5}$).

To understand how selective forces may be influencing the diversity observed within the pangenome, we further estimated $d_N/d_S$, the ratio of nonsynonymous to synonymous nucleotide mutations within every gene of the pangenome (Methods). Consistent with the low level of diversity observed, the average $d_N/d_S$ value was 0.28, and 96% of the

6,853 genes for which it was successfully estimated (Methods) had $d_N/d_S$ <1 (Fig. 4b). This implies that negative selection is widespread across the coding regions of the *H. influenzae* genome. Of the remaining 256 genes (4%) with a $d_N/d_S$ estimate, 45 had $d_N/d_S$ >2, indicating potential positive directional selection. A further analysis of these genes was undertaken using three statistical tests implemented in the HYPHY v.2.5.60 (Methods), and a few accessory genes possessed extremely strong evidence of selection, where at least two of the three statistical tests rejected the null hypothesis of neutral evolution (Methods). The genes involved included an unnamed gluconate transporter, the BrnT toxin protein and a third small protein of unknown function. The results of these analyses are illustrated in Extended Data Figs. 2 and 3 and are briefly summarized here. The unnamed gluconate transporter showed statistically significant results in all three HYPHY tests used, and these tests indicated a branch of the gene phylogeny containing eight isolates and a specific codon (185) in the protein alignment which have been positively selected for. This branch consists of seven Maela isolates and one isolate from elsewhere, which all possess a structural variant of the unnamed gluconate transporter with a large deletion of a transmembrane domain. The *brnT* toxin gene, the toxin from the BrnT/BrnA type II toxin–antitoxin system[15] also showed statistically significant results in all three HYPHY tests, which indicated that a branch of gene phylogeny consisting of two Maela isolates with a large deletion of an alpha helix had recently been subject to positive selection. Finally, the protein of unknown function showed a statistically significant result in two of the three HYPHY tests, identifying a glutamine–valine variable site, with the valine variant primarily associated with Maela isolates and the glutamine variant primarily associated with isolates from elsewhere. All three of these proteins correspond to low-frequency accessory genes which are globally distributed. Notably, the variants identified as under selection in the *brnT* toxin and the unnamed gluconate transporter are either unique or much more prevalent among Maela isolates, with

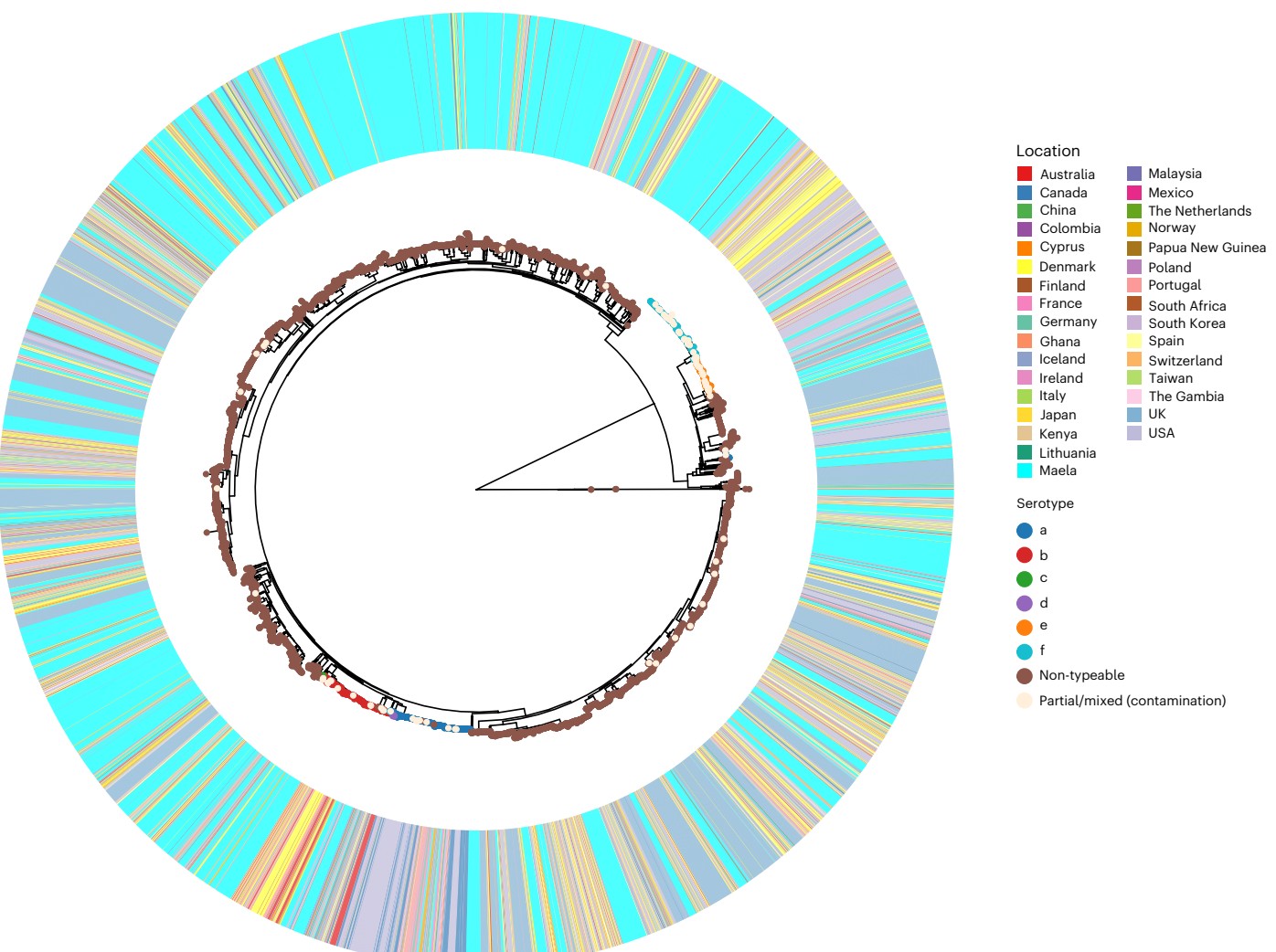

**Fig. 3 | Global *H. influenzae* phylogeny.** The maximum-likelihood core genome phylogeny of 9,849 *H. influenzae* isolates, combining the Maela cohort and a systematically identified global collection of published genomes. The phylogeny was estimated using IQ-TREE v.2.4.0 on the core genome alignment. The in silico serotypes are indicated by the circles on the tips of the phylogeny; the isolation location is indicated on the outer ring, as shown by colour as indicated in the legend. An interactive online phylogeny, with additional metadata including cgMLST, cgMLST clusters and partial disease state data, is available at ref. 85.

a similar split association between the two variants of the unnamed protein. This suggests that either the intensive longitudinal sampling frame or the circumstances of the Maela camp may be resulting in elevated statistical power to detect selection or genuinely stronger positive selection and rapid local adaptation.

Finally, to explore the geographic distribution of MDR lineages in greater detail, we focused on the lineages with ≥50 isolates and ≥30% resistance prevalence for at least four of nine antibiotic classes. This revealed that all large MDR lineages (*n* = 3) are widely disseminated internationally, that is, observed in at least 11 different locations (Fig. 5). One of these lineages was dominated by Hib strains and showed evidence of independent capsule switches to serotype a, whereas the rest were composed of NTHi.

## Discussion

The genomic epidemiology of non-b *H. influenzae* has remained largely elusive so far due to the lack of carriage studies in high-burden settings, particularly in populations from before the rollout of the Hib vaccine. Our study provides comprehensive evidence that NTHi are equally capable of causing invasive disease irrespective of their genetic background, even in a pre-Hib vaccine host population. Although only colonizing isolates were available from the Maela cohort, comparable sampling during episodes of clinical pneumonia has provided insights

into disease-associated strains. Results from the multi-country PERCH pneumonia aetiology study[8] confirmed a positive association between non-b *H. influenzae* upper respiratory colonization and chest X-ray confirmed pneumonia. The same study demonstrated an aetiologic fraction of 4.5% for non-b *H. influenzae* among human immunodeficiency virus-negative, chest X-ray-confirmed cases, which is comparable with the 6.7% fraction estimated for *Streptococcus pneumoniae*. There is evidence from several countries of an increasing burden of invasive non-b *H. influenzae* disease, notably in neonates and older adults, with the vast majority being NTHi infections[1,16]. The high burden of pneumonia attributable to NTHi and the notable childhood mortality associated with it (the third most common bacterial pathogen) in the low-resource settings in both Africa and Asia[16], combined with the frequent emergence of MDR lineages as identified in the current study, serve as a reminder of the health benefits of developing an immunization programme targeting the eradication of these pathogens. This would contribute not only towards removing the public health burden of non-B *H. influenzae* invasive disease, but also to substantially reduce both the incidence of AOM and the need to prescribe antibiotics to children.

A pre-vaccine carriage study conducted in The Gambia during the 1980s identified a highly variable carriage rate of serotype b, ranging between 0% and 33% across rural and urban areas, whereas the species-wide carriage rate was found to be 90% among children under

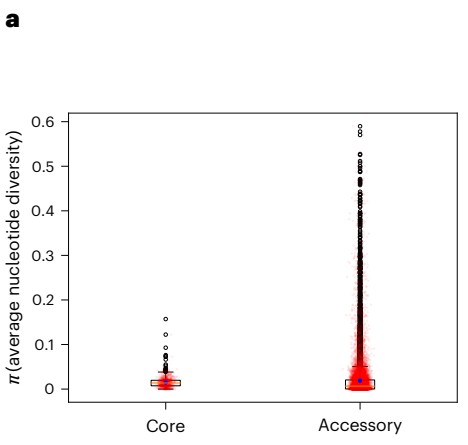

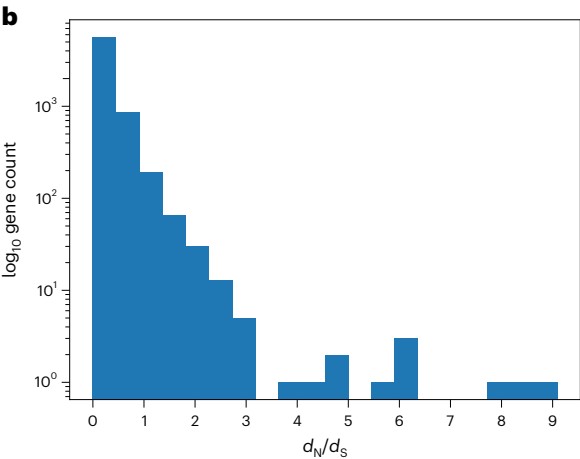

**Fig. 4 | Nucleotide diversity $d_N/d_S$ across the *H. influenzae* pangenome. a**, The box plots of the estimated average pairwise nucleotide diversity, $\pi$, in each gene of the aligned pangenome of the combined dataset, split into genes present in 80% or more isolates (core, $n = 1{,}044$) and genes in fewer than 80% of isolates (accessory, $n = 8{,}843$). The blue hexagons indicate gene-frequency weighted average nucleotide diversity across all genes; the yellow line is the median, the outer edges are the first and third quartiles, and the whiskers are 1.5× the interquartile range beyond those values. The black points indicate outliers, and all data points are plotted in transparent red. **b**, The log-scaled histogram of the estimated $d_N/d_S$ values, the ratio of nonsynonymous to synonymous mutations, across 6,853 aligned genes from the pangenome of the combined dataset.

5 years of age[17]. To our knowledge, there are no other comparable prevaccine studies with serotyping data, but the study in The Gambia suggests that the Maela *H. influenzae* are not atypical in terms of serotype distribution in an unvaccinated host population in a high-burden setting. Postvaccine carriage studies across Europe and China consistently show a decline of serotype b as expected but also that NTHi are most commonly colonizing young children and that all non-b serotypes are rare[18–21]. A Belgian study compared carriage rates among children attending day care and those diagnosed with either AOM or invasive disease during 2016–2018. Notably, NTHi were dominating in each category (colonizing, AOM, invasive), with the percentages 95.2%, 98.2% and 68.1%, respectively[19]. Similarly, in Norwegian (2017–2021) and Portuguese (2011–2018) national surveillance of *H. influenzae* invasive disease, NTHi accounted for 71.8% and 79.2% of the cases, respectively. These findings are well aligned with our data from Maela and further with a recent study of CAP in children under 5 years vaccinated against Hib in Vietnam, where a high fraction of NTHi was also detected using real-time PCR[22]. Interestingly, although serotype b has been found either completely absent[18,19] or very rare[20] in European carriage studies, it is still found in invasive disease across the continent[19,23,24], suggesting ongoing transmission from unvaccinated regions of the world.

Apart from the genomic epidemiology of *H. influenzae*, the overall understanding of the species' population structure has also remained largely elusive, particularly at a global scale, despite various efforts to elucidate it over the past decade. This study, through analysing a large cohort of isolates from an understudied region combined with a systematic collection of publicly available data, suggests that the global *H. influenzae* population is not structured into independently evolving lineages which are predominant in certain regions but rare in others. This is unlike other well-studied bacterial species that colonize the same niche, such as *S. pneumoniae* or *Neisseria meningitidis*, where distinct, independently evolving lineages have been readily identified for decades[25–27]. Based on our analyses, *H. influenzae*, in particular NTHi, instead appears to have a population structure reminiscent of panmixia, where routine gene flow between members of the species prevents the formation of stable lineages. This type of population structure would account for the limited success of various methods used to cluster the population in this study and the difficulties previous efforts have encountered when using smaller datasets[11]. Technically, clustering methods have probably been limited by the low levels of nucleotide diversity observed within the *H. influenzae* genome, as we

found no SNPs in over half the core genome of the Maela isolates. This, however, is not apparent in the output of clustering methods and only becomes evident when population genomics analyses are conducted at species-wide scale.

Despite the low levels of nucleotide diversity, the phylogenetic analysis of the combined Maela and globally sequenced isolates remains possible and clearly demonstrates in *H. influenzae* a persistent lack of phylogeographical signal (closely related isolates being highly colocalized), even with this collection of isolates spanning over 50 years. This strongly suggests that interregional and intercontinental transmission of these bacteria happens frequently. This is consistent with the high levels of recombination observed in the Maela dataset, as that would facilitate the efficient admixture of migrating isolates with the destination population. Furthermore, the frequent migration and recombination, when combined with widespread evidence of negative selection across coding regions of the genome implied by the low $d_N/d_S$ values, correspond to a pool of biological forces that probably explains the low nucleotide diversity of the *H. influenzae* genome. It is, however, difficult to disentangle the individual contributions of migration, recombination and negative selection in producing low levels of diversity, and indeed, they are probably acting in concert, as has been demonstrated previously in other ecological settings[28,29].

Our results underscore the importance of a global perspective on disease surveillance when developing public health strategies for managing invasive *H. influenzae* disease, as it is clear that pathogenic adaptations which arise in one part of the world have ample opportunity for global spread. Although the MDR lineages identified in this study were more frequent in Maela than any other individual sampling country, they were still mostly composed of isolates from around the world (55–65%). Due to the general bias of sampling towards high-income settings, we cannot exclude the possibility that these lineages may be further established in unsampled LMIC populations with high antibiotic use. Intensified efforts should thus be made to include *H. influenzae* into AMR surveillance programmes as widely as possible. Such surveillance should preferably not be limited to including only bloodstream isolates, because it will otherwise underestimate the prevalence of circulating AMR determinants among pneumonia and AOM clinical cases. Similar to *S. pneumoniae*, carefully conducted studies of *H. influenzae* colonization in pneumonia cases and controls may provide further data on the relative invasiveness of capsulated and unencapsulated strains[30]. Given the significant evidence of adaptation

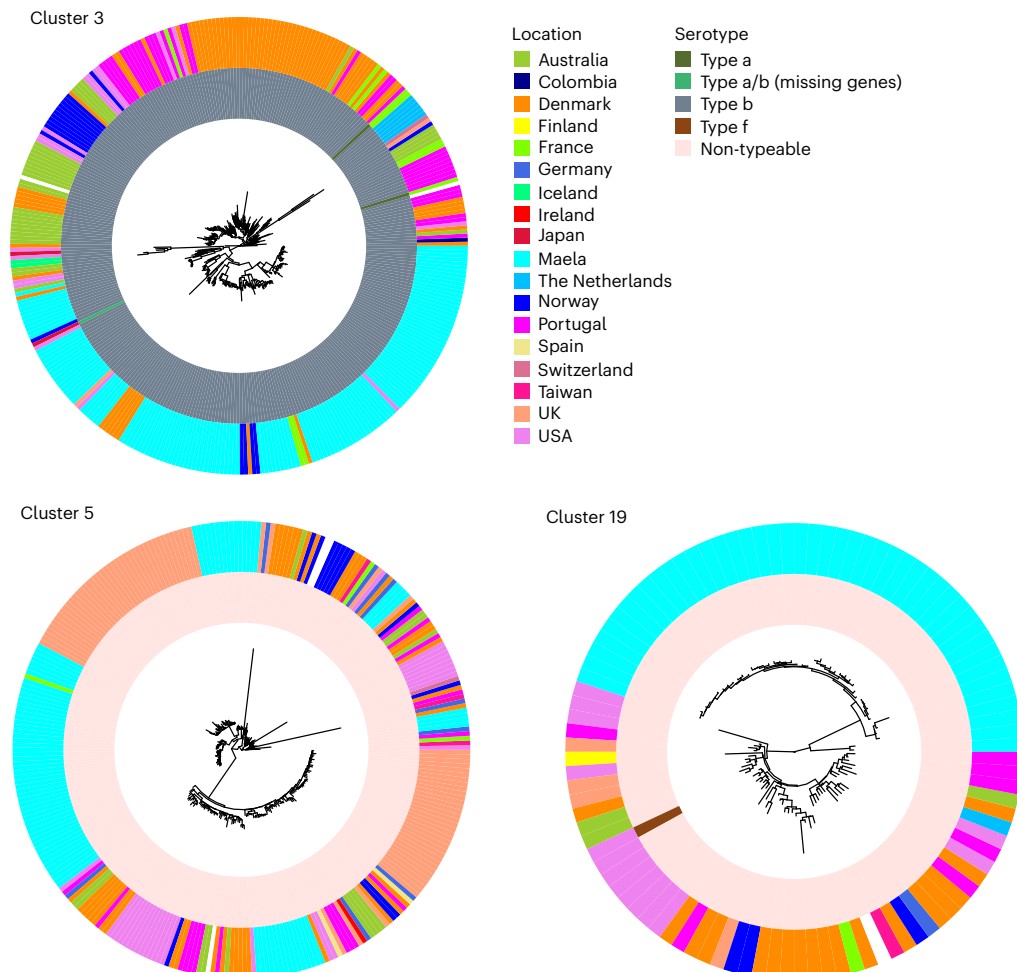

**Fig. 5 | Phylogenies of MDR *H. influenzae* lineages.** Recombination-free maximum-likelihood phylogenies for each PopPUNK MDR cluster. Each cluster comprises at least 50 genomes, with at least 30% resistance prevalence for a minimum of four of nine antibiotic classes (Fig. 2). The phylogenies were inferred by using Gubbins on whole-genome pseudoalignments of each cluster, separately mapped against the *H. influenzae* reference 86-028NP (NC_007146.2). The in silico serotypes (inner ring) and isolation location (second ring) are shown by colour as indicated in the legend. The isolates collected in Maela represent the most common origin in all three clusters (40% in cluster 3, 30% in cluster 5 and 45% in cluster 19).

in accessory genes in the Maela population and that the MDR lineages were predominantly identified among Maela isolates, it is possible that the camp host population may be exceptionally well suited to the evolutionary adaptation of these bacteria. This could be due to either the host population density resulting in high colonization and transmission success or the level of antibiotic use in the camp, and it is further feasible that the fitness cost of maintaining such high levels of resistance beyond these settings is prohibitive. An alternative and more plausible explanation is that the higher sampling density in the Maela cohort has led to higher statistical power to identify adaptation using methods based on aligned gene sequences, suggesting that similar adaptation could have also taken place elsewhere. The widespread genomic surveillance in comparable settings would facilitate early detection of the spread of extensive levels of AMR. Importantly, such surveillance data would also be crucial in developing a deeper understanding of how selection drives the evolution and maintenance of AMR in *H. influenzae*.

Apart from the concerning implication regarding the possibility of the global spread of AMR in *H. influenzae*, the results of this study also highlight the importance of vaccination against serotype b *H. influenzae*, where we have detected extremely high rates of MDR isolates. Our results further suggest that vaccination may be a particularly effective strategy to control invasive *H. influenzae* disease irrespectively of the serotype due to the lower level of diversity present within its core

genome relative to the accessory and its highly admixed population structure. Given the low level of the observed allelic diversity, the pervasive negative selection we detected throughout the *H. influenzae* genome at a global scale may be strong enough to overcome selection driving compensatory adaptations which would generally reduce the vaccine efficacy in response to rollout. Although this is a cause for optimism, it must be tempered by the fact that the high levels of recombination observed in *H. influenzae* may also increase the efficacy of positive selection on any mutations which do arise, as has been observed in other species[31]. In any case, the stark contrast between the *H. influenzae* population structure identified in this work and the highly stratified population structure of *S. pneumoniae*, both globally[27] and in the Maela host population[32], strongly suggests that vaccine evasion through interlineage competition and replacement, as has repeatedly been observed in *S. pneumoniae*[33], would be much less likely to happen in *H. influenzae*, due to the absence of a deeply structured population and local variants. A number of conserved surface antigens have been under investigation as potential candidates for protein subunit vaccines[34]. Recently, antigenic responses to several of the promising candidates have been measured for otitis-media-prone children and their controls; these include the recombinant soluble PilA (rsPilA) fused with protein E, protein D and the ubiquitous surface protein A2 (UspA2) from *Moraxella catarrhalis*, as well as ChimV4 (a chimera of protective epitopes from rsPilA) and the outer membrane protein P5

(OMP P5)[35]. Although there are many complications involved in the design of protein-based bacterial vaccines which would need to be overcome[36], our work supports the conjecture that a single universal vaccine could possibly be developed to combat invasive *H. influenzae* disease and suggests that the eradication of invasive disease caused by *H. influenzae* may be a feasible end goal of widespread vaccination campaigns.

## Methods

### Ethical approval
Written informed consent was obtained from the participating infants' mothers before enrolment into the cohort study. Ethical approval was granted by the ethics committees of the Faculty of Tropical Medicine, Mahidol University, Thailand (MUTM-2009-306) and Oxford University, UK (OXTREC-031-06). The sequencing work on stored isolates described here was approved by the same committees (TMEC-19-043; OxTREC-551-19).

### Study design and collections
A total of 4,474 *H. influenzae* isolates were retrieved from a mother–infant cohort of 999 pregnant women from the Maela camp for displaced persons, Thailand, from October 2007 to November 2008[37,38]. Within a 24-month postpartum period, the infants were sampled by NPSs monthly and when the infant presented symptoms of World Health Organization (WHO) clinical pneumonia. For comparative analyses, a systematic search was conducted for publicly available short-read genome sequences for which country and year of isolation metadata was available. The data were retrieved from the ENA for 6,129 isolates[9,10,13,39–66], of which 5,879 passed QC, resulting in a final dataset size of 9,849 isolates.

### Sampling and sequencing procedures
Between October 2007 and November 2008, 999 pregnant women from the Maela camp for displaced persons (located on the Thailand–Myanmar border in Tak province, northwest Thailand (Fig. 1a,b)) were recruited into a mother–infant colonization study. The infants were followed from birth for 24 months, and a nasopharyngeal swab (NPS) specimen was collected (dacron tipped swabs; Medical Wire & Equipment) at monthly intervals and if the infant presented to the Shoklo Malaria Research Unit clinic with symptoms and signs compatible with WHO clinical pneumonia (Fig. 1b).

Following sampling, the NPS tip was excised immediately into a sterile cryovial containing 1 ml skim milk, tryptone, glucose, glycerol medium (STGG; prepared in-house) using 70% ethanol-cleaned scissors. The NPS–STGG specimens were transferred to the Shoklo Malaria Research Unit (SMRU) microbiology laboratory in a cool box within 8 h of collection and were frozen at −80 °C until culture.

In total, 10 µl of thawed NPS–STGG specimen was cultured onto plain chocolate agar (Clinical Diagnostics), a 10-unit bacitracin disc (Oxoid) applied to the first streak, and the plate was incubated overnight at 36 °C in 5% CO$_2$. The bacitracin-resistant colonies were confirmed as *H. influenzae* by Gram stain and X + V factor-dependent growth. The serotype was determined by slide agglutination (Becton Dickinson) and by in silico capsule typing[12]. Two isolates, one NTHi and one serogroup b by agglutination, gave only partial in silico capsule typing[12] results. The in silico capsule typing was generally congruent (overall congruence 95.3%) with the agglutination-based phenotypic serotyping, and was corrected by the latter in those 28 cases where the serological typing indicated a serotype for a NT in silico type. The pure isolates of *H. influenzae* were collected from an overnight culture plate into 1 ml of STGG and stored at −80 °C before DNA extraction at Qiagen using the DNeasy protocol.

Short-read WGS of the 4,474 *H. influenzae* isolates was performed at the Sanger Wellcome Institute on the Illumina-HTP NovaSeq 6000 platform with 150-bp paired-end sequencing.

For long-read sequencing, one reference isolate was selected per each of the 48 largest PopPUNK clusters, covering 3,558 (89.6%) of 3,970 isolates of the study cohort. The reference isolates were selected on the basis of the gene presence absence matrix from the estimated pangenome, using a published selection pipeline[67,68].

The selected *H. influenzae* strains were subcultured on chocolate agar and incubated overnight at 35–37 °C in 5% CO$_2$. The genomic DNA was extracted using the Qiagen MagAttract HMW DNA Kit. The WGS libraries were constructed using the Oxford Nanopore Technologies (ONT) SQK-NBD112.96 Native Barcoding Kit, and all 48 strains were pooled together and sequenced on one ONT R9.4.1 flowcell using a MinION Mk1c. The hybrid assembly of the reference isolates was performed using a publicly available pipeline[69] with a minimum ONT coverage of 40× and a phred score of 20 to trim the Illumina reads, resulting in 40 (83.3%) complete hybrid assemblies of the 48 reference isolates.

### Genomic Analysis
A total of 4,474 *H. influenzae* isolates were sequenced at the Wellcome Sanger Institute on NovaSeq 6000 150-bp paired-end platform. Species contamination was identified by using Kraken v.0.10.6 (ref. [70]), and the sequence data failed QC if the depth of coverage was <20× or if there was evidence of contamination or mixed strains, poor assembly or extreme violation of any of the QC parameters. In total, 3,970 isolates passed QC and were included in the genomic analyses. Short-read genome sequences, both newly sequenced and publicly available, were assembled and annotated using a published pipeline with default parameters[71], and QC on all isolates was performed on the basis of the number of contigs, genes and distance from the origin in an multidimensional scaling projection of all pairwise distances. The Maela isolates were clustered using PopPUNK v.2.4.0[72] with a core threshold of 0.11, whereas the default threshold was used for the combined global data. Antimicrobial resistance genes and point mutations were screened from assemblies using AMRFinderPlus v.4.0.3 and *H. influenzae*-curated database v.2024-12-18.1 (--organism Haemophilus_influenzae) with a minimum identity of 75% and a minimum coverage of 80%. Hicap v.1.0.3[12] was used to infer capsule type from assemblies. cgMLST was identified using chewBBACA[73] and the *H. influenzae* cgMLST database[11], and a simple network clustering method was used to group isolates into complexes on the basis of the number of mismatches in their allele profiles (either 100 or 250). For the phylogenetic analyses on the Maela data, the sequence reads of the 3,970 genomes were mapped to the complete genome of *H. influenzae* 86-028NP (NC_007146.2) (ref. [74]) using Snippy v.4.6.0[75], and an SNP-only sequence alignment was created using snp-sites v.2.5.1[76]. A phylogeny for Maela collection was inferred using FastTree v.2.1.10 with a generalized time-reversible model[77], and a maximum-likelihood tree was inferred for the global collection using IQ-TREE v.2.4.0[78,79] on Panaroo core-genome alignment, with uninformative regions masked using information entropy scores.

The pangenome was inferred for the Maela genome collection using Panaroo v.1.2.9[80] using sensitive mode and merging paralogs. The pangenome was further inferred for the entire combined global dataset running Panaroo in strict mode. FastGEAR v.2016-12-16 was used to infer recombinations, and pixy v.1.2.7.beta1 was used to infer per-gene nucleotide diversity, $\pi$, both from aligned pangenome gene sequences. Genomegamap v.1.0.1[81] was used to infer the maximum-likelihood estimates of the average $d_N/d_S$ for each gene in the pangenome. Only genes with an estimated nucleotide diversity ($\theta$) >0.005 were considered robust enough estimates for further interpretation ($n = 6,853$ genes). Genes with robust estimates of $d_N/d_S$ greater than 2 were further analysed using the HYPHY package v.2.5.60[82], specifically the FUBAR, FEL and ABSREL statistical tests for pervasive gene-wide directional selection, specific sites subject to directional selection and subsets of branches subject to directional selection respectively. Genes which had significant results to at least one of these HYPHY

tests were manually investigated by searching the consensus and variant nucleotide sequences against the non-redundant protein database with tblastx, searching ESMFold-predicted protein structures against the AlphaFold, Uniprot and Swiss-prot database using Foldseek. Three-dimensional structural alignments of consensus, variant and reference protein structures were created for specific genes using TM-align.

MDR clusters in the combined collection were defined as PopPUNK clusters with ≥50 isolates, of which ≥30% harboured resistance determinants to greater than or equal to four out of nine antibiotic classes (aminoglycoside, β-lactam, phenicol, sulfonamide, tetracycline, trimethoprim, macrolide, quinolone and rifamycin). For the phylogenetic reconstruction of the global MDR clusters, assemblies of each cluster were mapped to the reference *H. influenzae* 86-028NP[74] using Snippy, and phylogenies were inferred using Gubbins[83].

To test for an association between the lineages of NTHi and invasive disease, we deduplicated isolates from the same host likely to be clonally related to remove the bias associated with longitudinal sampling. The deduplication was performed by first grouping all isolates sampled from the same hosts within 60 days that belonged to the same PopPUNK cluster and then randomly selecting a single isolate for the association analysis. The deduplicated NTHi isolates spanned 120 PopPUNK clusters of the Maela cohort, and we performed a permutation test of association between these and pneumonia by randomly shuffling sample labels (pneumonia and non-pneumonia). A two-sided Fisher's exact test with 10,000 Monte Carlo replications was used.

### Reporting summary

Further information on research design is available in the Nature Portfolio Reporting Summary linked to this article.

## Data availability

All sequence data generated as part of this Article are available via ENA/SRA/DDBJ under study accession no. PRJEB41043. The metadata for both newly sequenced data and the systematic global collection are provided in the Supplementary Information and are available via Microreact for interactive exploration as indicated in the figure captions[84,85]. The sequence data were produced according to Illumina protocols, and publicly available data were downloaded from the ENA using enadownloader.

## Code availability

No algorithms or computational methods were developed during the course of this work.

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

## Acknowledgements

This work was supported by European Research Council (grant no. 742157 to J.C.), Wellcome Trust Grants (grant nos. 206194 and 083735 to P.T.; MORU core award no. 220211 to P.T.; grant nos. 206194 and 220540/Z/20/A to Wellcome Sanger Institute), Trond Mohn Foundation (BATTALION grant to J.C., A.K.P., S.M. and N.M.) and Marie Skłodowska-Curie Actions (grant no. 801133 to A.K.P., S.M. and N.M.).

## Author contributions

N.M. and A.K.P. had the major responsibility in bioinformatics, population genomics and statistical analyses. C.L. was responsible for the management and provision of swab specimens and isolates and the curation of isolate data. S.M. conducted additional statistical analyses. J.T. was responsible for the investigation of isolates. F.H.N. administered the cohort study. C.T. investigated and curated cohort clinical data. S.D.B. advised on the study design and interpretation of results. N.J.C. advised on population genomics and interpretation of results. P.T. and J.C. acquired funding and jointly designed and supervised the study. N.M., A.K.P., P.T. and J.C. jointly wrote the initial draft and all other authors improved the paper.

## Competing interests

The authors declare no competing interests.

## Additional information

**Extended data** is available for this paper at https://doi.org/10.1038/s41564-025-02171-9.

**Correspondence and requests for materials** should be addressed to Paul Turner or Jukka Corander.

[1]Department of Biostatistics, University of Oslo, Oslo, Norway. [2]Parasites and Microbes, Wellcome Sanger Institute, Hinxton, UK. [3]Cambodia Oxford Medical Research Unit, Angkor Hospital for Children, Siem Reap, Cambodia. [4]Centre for Tropical Medicine and Global Health, Nuffield Department of Medicine, University of Oxford, Oxford, UK. [5]Peter MacCallum Cancer Centre, Melbourne, Victoria, Australia. [6]Sir Peter MacCallum Department of Oncology, University of Melbourne, Melbourne, Victoria, Australia. [7]Mahidol-Oxford Tropical Medicine Research Unit, Faculty of Tropical Medicine, Mahidol University, Bangkok, Thailand. [8]Shoklo Malaria Research Unit, Mahidol-Oxford Tropical Medicine Research Unit, Faculty of Tropical Medicine, Mahidol University, Mae Sot, Thailand. [9]MRC Centre for Global Infectious Disease Analysis, Department of Infectious Disease Epidemiology, Imperial College London, London, UK. [10]Department of Mathematics and Statistics, University of Helsinki, Helsinki, Finland. [11]These authors contributed equally: Neil MacAlasdair, Anna K. Pöntinen, Clare Ling. [12]These authors jointly supervised this work: Paul Turner, Jukka Corander. ✉e-mail: pault@tropmedres.ac; jukka.corander@medisin.uio.no

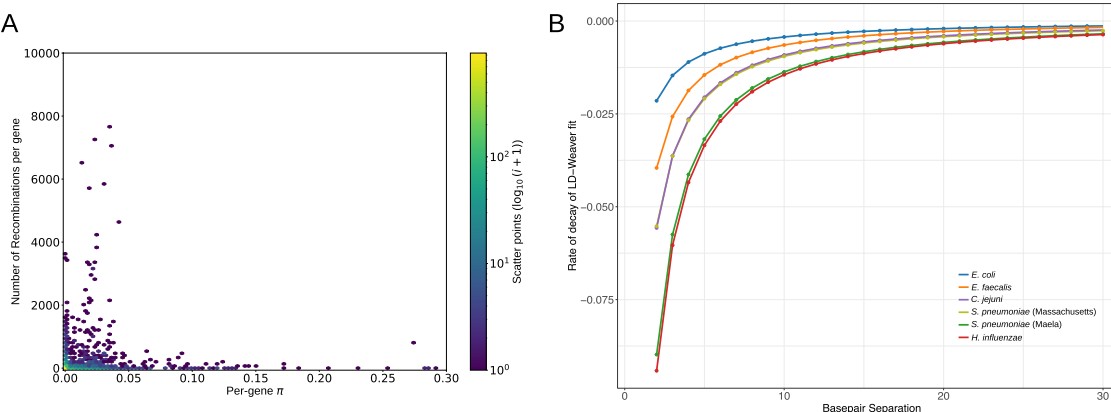

**Extended Data Fig. 1 | Associations between estimated nucleotide diversity and recombination, and between decay in linkage disequilibrium and genomic distance.** Panel **A**: Log-scaled hexagon-density scatter plot of the estimated per-gene nucleotide diversity (π), versus the number of recombination events inferred per-gene. Hexagon colours indicate the number of scatter plot points (log-scaled) present within each hexagon. Panel **B**: The estimated rate of decay in linkage disequilibrium (LD) for the *H. influenzae* Maela cohort, compared with estimated rates for the four species with LD decay functions fitted in the reference. [14]. The shown curves correspond to gradients of the LD decay function, with smaller values indicating faster decoupling of SNPs as a function of distance in base pairs.

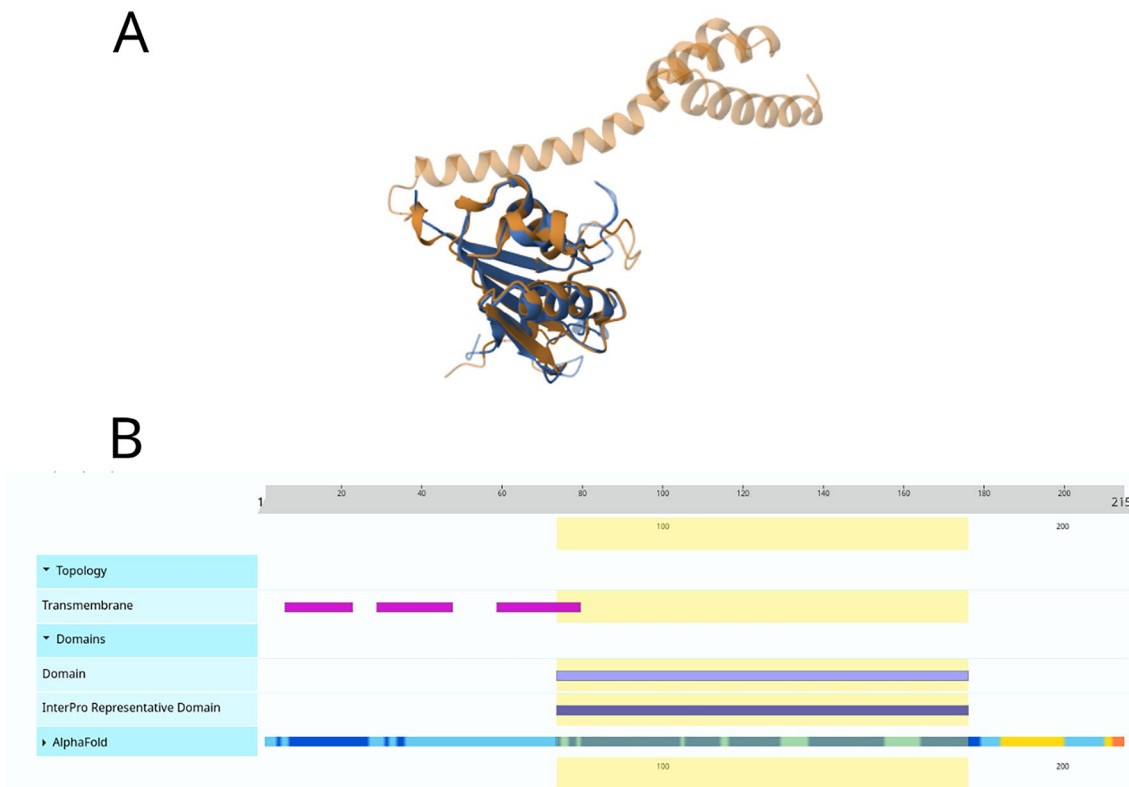

**Extended Data Fig. 2 | Unnamed Gluconate Transporter (pangenome ID: group_3361) Identified as being under selection in some Maela isolates. A**: Structural alignment of variant under selection in Maela data (Blue) and wild type variant (Gold), with the large domain composed of alpha helices at the C-terminus deleted in the Maela variant. **B**: Interpro scan domain annotation of the wild-type protein indicating that the deleted C-terminus domain in the Maela data is transmembrane.

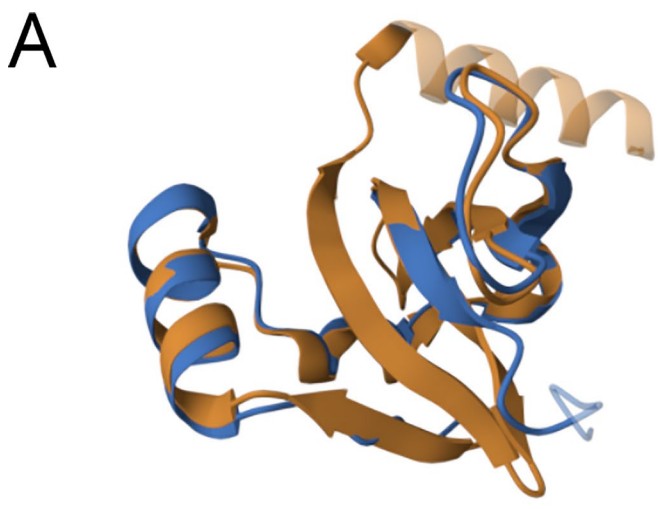

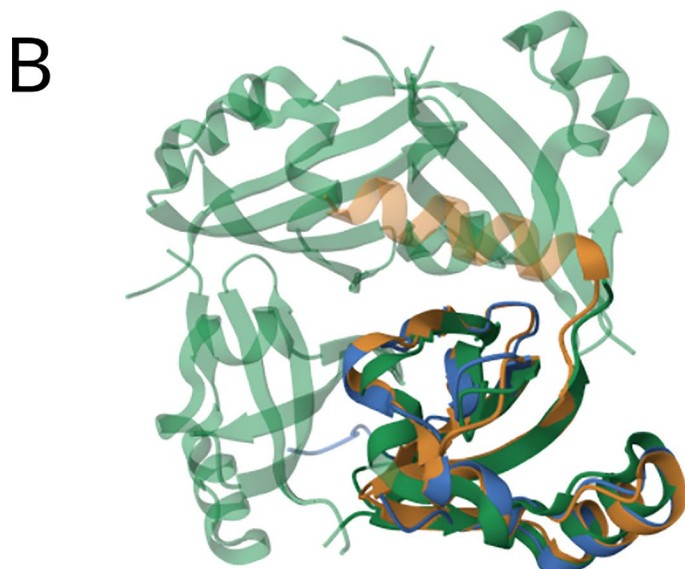

**Extended Data Fig. 3 | BrnT toxin (pangenome ID: group_4059) variant includes an altered alpha helix domain. A**: Structural alignment of the structural variant found in Maela isolates (blue) compared with the wild type *H. influenzae* structure (gold). The variant under selection contains some amino acid changes, as well as a small deletion, leading to the absence of an alpha helix when compared to the wild type *H. influenzae* structure. **B**: Structural alignment of both *H. influenzae* BrnT genes compared to the reference structure from *Brucella abortus*.

# Reporting Summary

## Statistics

For all statistical analyses, confirm that the following items are present in the figure legend, table legend, main text, or Methods section.

| n/a | Confirmed | |
|---|---|---|
| ☐ | ☒ | The exact sample size (*n*) for each experimental group/condition, given as a discrete number and unit of measurement |
| ☐ | ☒ | A statement on whether measurements were taken from distinct samples or whether the same sample was measured repeatedly |
| ☐ | ☒ | The statistical test(s) used AND whether they are one- or two-sided<br>*Only common tests should be described solely by name; describe more complex techniques in the Methods section.* |
| ☒ | ☐ | A description of all covariates tested |
| ☒ | ☐ | A description of any assumptions or corrections, such as tests of normality and adjustment for multiple comparisons |
| ☐ | ☒ | A full description of the statistical parameters including central tendency (e.g. means) or other basic estimates (e.g. regression coefficient) AND variation (e.g. standard deviation) or associated estimates of uncertainty (e.g. confidence intervals) |
| ☐ | ☒ | For null hypothesis testing, the test statistic (e.g. *F*, *t*, *r*) with confidence intervals, effect sizes, degrees of freedom and *P* value noted<br>*Give P values as exact values whenever suitable.* |
| ☒ | ☐ | For Bayesian analysis, information on the choice of priors and Markov chain Monte Carlo settings |
| ☒ | ☐ | For hierarchical and complex designs, identification of the appropriate level for tests and full reporting of outcomes |
| ☒ | ☐ | Estimates of effect sizes (e.g. Cohen's *d*, Pearson's *r*), indicating how they were calculated |

*Our web collection on statistics for biologists contains articles on many of the points above.*

## Software and code

Policy information about availability of computer code

| Data collection | Sequence data was produced according to Illumina protocols, and publicly available data was downloaded from the ENA using enadownloader. |
|---|---|
| Data analysis | No novel algorithms or computational methods were developed during the course of this work. |

For manuscripts utilizing custom algorithms or software that are central to the research but not yet described in published literature, software must be made available to editors and reviewers. We strongly encourage code deposition in a community repository (e.g. GitHub). See the Nature Portfolio guidelines for submitting code & software for further information.

## Data

Policy information about availability of data

All manuscripts must include a data availability statement. This statement should provide the following information, where applicable:

- Accession codes, unique identifiers, or web links for publicly available datasets
- A description of any restrictions on data availability
- For clinical datasets or third party data, please ensure that the statement adheres to our policy

All sequence data generated as part of this work is available on the ENA/SRA/DDBJ under study accession PRJEB41043. Metadata for both newly sequenced data and the systematic global collection is available as Supplementary Information and on microreact at the links as indicated in figure captions. (https://microreact.org/project/oMm8PFCoG2429JwiDBpdru-maela-h-influenzae and https://microreact.org/project/ioyt4oJRSJgeFGK9KmFyVk-global-h-influenzae-core-tree)

# Research involving human participants, their data, or biological material

Policy information about studies with human participants or human data. See also policy information about sex, gender (identity/presentation), and sexual orientation and race, ethnicity and racism.

| | |
|---|---|
| Reporting on sex and gender | The original Maela cohort study recorded biological sex of infants, as determined by routine physical examination by the attending midwife shortly after birth. The written informed consent process did not explicitly cover reporting of sex in downstream publication. Data on the sex of infants is not reported in the current manuscript. |
| Reporting on race, ethnicity, or other socially relevant groupings | Maela cohort study participants were from Myanmar and predominantly of Karen ethnicity. Apart from describing the geographic location of Maela camp for displaced persons, the manuscript does not include any race, ethnicity or social group data. |
| Population characteristics | Maela cohort study participants were displaced persons from Myanmar, residing in Maela camp (northwest Thailand). |
| Recruitment | Between October 2007 and November 2008, all pregnant women attending the Shoklo Malaria Research Unit (SMRU) antenatal clinic for pregnancy care were invited to consent to their infant's participation in a pneumonia / pneumococcal carriage cohort study. There were no exclusion criteria. The SMRU clinic was the sole provider of antenatal care for Maela camp at that time. |
| Ethics oversight | Written informed consent was obtained from the participating infants' mothers prior to enrolment into the cohort study. Ethical approval was granted by the ethics committees of the Faculty of Tropical Medicine, Mahidol University, Thailand (MUTM-2009-306) and Oxford University, UK (OXTREC-031-06). The sequencing work on stored isolates described here was approved by the same committees (TMEC-19-043; OxTREC-551-19). |

Note that full information on the approval of the study protocol must also be provided in the manuscript.

# Field-specific reporting

Please select the one below that is the best fit for your research. If you are not sure, read the appropriate sections before making your selection.

☒ Life sciences ☐ Behavioural & social sciences ☐ Ecological, evolutionary & environmental sciences

For a reference copy of the document with all sections, see nature.com/documents/nr-reporting-summary-flat.pdf

# Life sciences study design

All studies must disclose on these points even when the disclosure is negative.

| | |
|---|---|
| Sample size | As our aim was to study the entire population of H. influenzae in the refugee camp (disease and carriage) we collected as many samples as feasible during routine intervals. |
| Data exclusions | Sequence data were excluded on the basis of several quality control metrics, including number of sequenced reads, assembly N50, reference genome mapping coverage or presence of contaminant sequences. |
| Replication | As this is a large whole-genome sequencing study, it is not possible to replicate without repeating the entire study in a novel context. |
| Randomization | As this is not a comparative study, there are no experimental groups. |
| Blinding | Investigators analysing the whole-genome sequence data were not involved in the sampling, culturing, or sequencing of any initial biological data. |

# Reporting for specific materials, systems and methods

We require information from authors about some types of materials, experimental systems and methods used in many studies. Here, indicate whether each material, system or method listed is relevant to your study. If you are not sure if a list item applies to your research, read the appropriate section before selecting a response.

## Materials & experimental systems

| n/a | Involved in the study |
|-----|-----------------------|
| ☒ | Antibodies |
| ☒ | Eukaryotic cell lines |
| ☒ | Palaeontology and archaeology |
| ☒ | Animals and other organisms |
| ☒ | Clinical data |
| ☒ | Dual use research of concern |
| ☒ | Plants |

## Methods

| n/a | Involved in the study |
|-----|-----------------------|
| ☒ | ChIP-seq |
| ☒ | Flow cytometry |
| ☒ | MRI-based neuroimaging |

## Plants

| | |
|---|---|
| Seed stocks | *Report on the source of all seed stocks or other plant material used. If applicable, state the seed stock centre and catalogue number. If plant specimens were collected from the field, describe the collection location, date and sampling procedures.* |
| Novel plant genotypes | *Describe the methods by which all novel plant genotypes were produced. This includes those generated by transgenic approaches, gene editing, chemical/radiation-based mutagenesis and hybridization. For transgenic lines, describe the transformation method, the number of independent lines analyzed and the generation upon which experiments were performed. For gene-edited lines, describe the editor used, the endogenous sequence targeted for editing, the targeting guide RNA sequence (if applicable) and how the editor was applied.* |
| Authentication | *Describe any authentication procedures for each seed stock used or novel genotype generated. Describe any experiments used to assess the effect of a mutation and, where applicable, how potential secondary effects (e.g. second site T-DNA insertions, mosiacism, off-target gene editing) were examined.* |

