## [Peer Review File · Nature Microbiology]

Genetic population structure of *Haemophilus influenzae* at local and global scales

Corresponding Author: Professor Jukka Corander

Version 0:

Decision Letter:

25th October 2024

Dear Professor Corander,

Thank you for submitting your Article entitled "The major pathogen *Haemophilus influenzae* experiences pervasive recombination and purifying selection at local and global scales" for consideration and please accept our apologies for the time it has taken us to contact you with a decision on your manuscript, which is due to our current high submission volume. We are very interested in your study, but we noticed that the manuscript currently lacks a section on ethics in the methods. Before we can make a final decision, please resubmit a revised manuscript with ethical information regarding the human cohorts.

Link Redacted

Nature Microbiology is committed to improving transparency in authorship. As part of our efforts in this direction, we are now requesting that all authors identified as 'corresponding author' on published papers create and link their Open Researcher and Contributor Identifier (ORCID) with their account on the Manuscript Tracking System (MTS), prior to acceptance. This applies to primary research papers only. ORCID helps the scientific community achieve unambiguous attribution of all scholarly contributions. You can create and link your ORCID from the home page of the MTS by clicking on 'Modify my Springer Nature account'. For more information please visit www.springernature.com/orcid.

Yours sincerely,

Version 1:

Reviewer comments:

Reviewer #2

(Remarks to the Author)

General comments

While this study provides useful information using whole genome sequence data from isolates of *Haemophilus influenzae* collected from the Mae La camp for displaced persons in Northwestern Thailand and similar WGS data publicly available, there are parts of the manuscript that can be improved for clarity. Although it has been stated in the "Introduction" (lines 82-84) "As a consequence, the genetic population structure and evolutionary dynamics of the species remain poorly understood, particularly at a global scale", the objectives or aims of this study has not been clearly spelled out. Have the authors used their data to answer these questions, (a) the genetic population structure and (b) evolutionary dynamics of *H. influenzae*?

Also since the longitudinally collected samples of *H. influenzae* is likely to be different from *H. influenzae* population with publicly available WGS data (time of collection of isolates, geographical origin of the isolates, and social economic background of the host that yielded these isolates), the WGS data from these two apparently very different populations should be analyzed separately and their results compared rather than to look at as one population. By lumping the analysis together may potentially blur any findings of significance if it involves small number of isolates especially for isolates collected from the later host population.

Another important but not considered carefully by the authors is the fact that the genomic epidemiology of serotype *H. influenzae* is very different from that of non-typeable *H. influenzae*. Therefore, general statement like this "*H. influenzae* appears to have a population structure reminiscent of panmixia, where routine gene flow between members of the species prevents the formation of

stable lineages” should be qualified by whether this applies to serotypeable or non-typeable H. influenzae as the former is monophyletic and the later is paraphyletic. I would suggest the authors to go over the whole manuscript and to make the distinction between serotype and non-typeable H. influenzae wherever this applies.

Some statements may need to be qualified when quantitative terms are applied. For example, this statement “Technically, clustering methods have likely been limited by the low levels of nucleotide diversity observed within the H. influenzae genome, particularly the core genome, even at a global scale” when referring to “low levels of nucleotide diversity, particularly the core genome” is somewhat puzzling since overall, the amount of genetic diversity is large as reflected in the number of unique MLST allelic profiles found in the Maelae samples (2344 isolates with allelic profiles not found in the MLST database and it is likely many of them are (a) non-typeable strains and (b) not of the same profile, although the exact serotype nature of the isolates yielding these new allelic profile and the exact number of unique allelic profile were not described by the authors. This is also evident in the large number of allelic sequences found in the H. influenzae MLST database (412,757 allele sequences from 10,999 isolates when the database was accessed on Dec 29, 2024).

Also, the description of a highly recombinant species (lines 166-167) and the low level of nucleotide diversity (line 286) needs to be addressed/discussed with a more comprehensive view.

Various terms were used to address the genetic or population structure of H. influenzae, such as clade (line 105), lineages (line 125) and clusters (line 128). Are these terms used consistently? Or defined in the manuscript.

Finally, it is unclear what is the conclusion of this study and does the conclusion address the objectives of this study.

Specific comments

- Line 38, in abstract, >4000, is it possible to provide the exact number here?
- Line 40-42, the pan-resistant lineages, were they NT or of all the Hi regardless of serotype?
- Table 1, was the data on serotype distribution based on in-silico from WGS data or agglutination based phenotype serotyping? Please add this information to the title of the Table.
- Line 106-110: accuracy of the in-silico capsule typing for serotype d and serotype e may not reflect the actual serotype nature of the isolates involved, there were three isolates identified by serotyping which means an error of either $3/8 = 37.5\%$ or $3/11 = 27.3\%$ for serotype d and for serotype e, the error maybe either $5/36 = 13.9\%$ or $5/41 = 12.3\%$. Therefore, the statement on lines 106-107, “In silico capsule typing was generally congruent with the agglutination-based phenotypic serotyping” should be qualified by adding the following, “except serotype d and serotype e”.
- Line 106, “Two isolates gave only partial in silico capsule typing results”, what were the serotyping results of these two isolates (which were included in Figure 2). Why is this information on serotype by phenotypic agglutination test not added in here?
- Line 122, please provide % for 2344/3970 of isolates; this means 58.9% of the H. influenzae isolates from Maela were made up of new sequence types not previously described elsewhere.
- Page 128, what is meant by “optimal clustering”?
- Page 127-131, does it mean PopPUNK grouped the 3970 isolates into 122 clusters but only 13 clusters contained at least 100 isolates and the largest cluster contained 349 isolates while 50% of the clusters (which means half of 122 clusters or 61 clusters) contained 10 or fewer isolates?
- Lines 129-131, the statement “with only 13 PopPUNK clusters in total each representing at least 100 isolates (out of a total of 122 clusters)”, should it be consistent to describe or add that there were 20 clusters with >50 isolates? (as described in the legend for Figure 2)
- Lines 124-127: the data presented in Supplementary Figure 1 would be more informative and easier to read if presented as a table involving the columns of Clonal complex, Sequence Type, Serotype, and number of isolates per category. Right now it is difficult to tell if the paraphyletic lineages involved serotypeable isolates or non-typeable isolates. Or keep Supplementary Figure 1 but also added the data as a supplementary Table.
- Line 138-141. It appears more isolates of Hib carried AMR determinants than NT-Hi (please describe % of Hib with AMR determinants versus % of NT-Hi with AMR determinants; also % of invasive Hib with AMR determinants versus % of carriage Hib with AMR determinants; % of invasive NT-Hi with AMR determinants versus % of carriage NT-Hi with AMR determinants; % of invasive H. influenzae with AMR determinants versus % of carriage H. influenzae with AMR determinants; an even better is to present this data by serotype (a, b, e, f, and non-typeable).
- Line 179-185: It is unclear if the largest cluster (composing of 483 isolates) was non-typeable or serotypeable? And the serotype nature of the smaller cluster (with 10 or fewer isolates) was also not described. These should be added here for completeness.
- Lines 180-181, regarding this statement “largest cluster composed of 483 isolates, and 19 clusters containing at least 100 isolates”, what serotype is the largest cluster? And is it monophyletic or paraphyletic?
- Figure 3, the in-silico serotype would be easily to read if the were labelled by a letter instead of colour (since the colour legend was very difficult to read) especially if serotypes were usually restricted by their clustering within the phylogeny.
- Lines 196-197, is this statement correct? “extremely low level of nucleotide diversity evident during the recombination analysis of the Maela cohort”, in view of this earlier statement, lines 166-167, “H. influenzae population within Maela is extremely Recombinant”?
- Lines 315-316, please refer to the data presented in the “Results Section” to support this statement “all but one of the pan-resistant MDR lineages was predominantly identified in the newly sequenced Maela isolates,” (is this figure 6 and would need some elaboration).

Reviewer #3

(Remarks to the Author)

In this study, MacAlasdair and co-authors sequenced and analysed over 4,000 isolates of H. influenzae from an unvaccinated paediatric carriage and pneumonia cohort. This is the first time that a large collection of both carriage and disease isolates of the

same population have been sequenced in a defined geographical region. Among the main findings, the authors found that serotype b was rarely found, with 91.7% of isolates being unencapsulated. They also found that the ability to cause invasive disease is not restricted to any subpopulation.

Major comments:

- An important observation of this study is that “non typable isolates made up 91.7% of all isolates, and serotype b isolates are the second most prevalent, making up 5.7% of the population.”. How generalisable is this observation? Can the authors contextualise this observation in the Discussion? are these proportions of NTHi and Hib similar in carriage and infection Hi of other strain populations?
- It seems numbers and proportions of H. influenzae isolates reported in the manuscript are indicated for the overall number of H. influenzae isolates that passed QC (denominator) but considering the dataset includes multiple longitudinal isolates per infant (“isolates were retrieved from a mother-infant cohort of 999 pregnant women from the Maela”) the authors should also provide figures and proportions of H. influenzae isolates in the host deduplicated dataset.
- In Results section “Genetic population structure of H. influenzae in Maela”: the authors conclude that “MLST sequence types (STs) and clonal complexes (CCs) [...] were unable to provide meaningful insight into the population structure.” because the “majority of the Maela isolates (2,344/3,970) contained combinations of alleles not present in the current MLST database”. This statement is misleading, the authors also state that STs and CCs do split the population into groups of closely related isolates, therefore the STs and CCs indeed provide some information on population structure. The lack of previously reported MLST profiles in the study population does not necessarily compromise the applicability of MLST typing more broadly. The identification of paraphyletic lineages of STs and CCs, interspersed with isolates from other STs/CCs, is instead what limits the applicability of MSLT in H. influenzae. Please rephrase these statements to make this point clearer. See my next comment on the use of cgMLST.
- The authors should attempt to build a core-genome phylogeny using a robust ML-based phylogenetics tool such as IQTree, using the genetic variation in the genes of the recently developed core genome multilocus sequence typing scheme for H. influenzae: <https://www.microbiologyresearch.org/content/journal/mgen/10.1099/mgen.0.001281>
Can the authors observe a better correlation between PopPUNK clusters and core genome clusters?
- The Results section ‘Distribution of AMR determinants’ could be extended to report on the prevalence and distribution of acquired AMR mutations and genes in the phylogeny. Also, as far as I am aware, AMRFinderPlus does not support the detection of AMR specifically for H. influenzae, which may miss important AMR mutations. This limitation should be mentioned and discussed. The authors may want to consider the recent work of Matthias Merker’s group in detecting AMR from Haemophilus influenzae genomes (e.g., <https://pubmed.ncbi.nlm.nih.gov/35139905/> and <https://pubmed.ncbi.nlm.nih.gov/39633433/>). Also, in the microreact project metadata, include the specific AMR genetic determinants identified by AMRFinderPlus, not just if isolates have (‘Yes’) or not (‘No’) such determinants.
- In the Results section ‘Quantification of homologous recombination’ the authors quantified the extended of homologous recombination per gene across all genes in the pan-genome of H. influenzae. The authors found that 38% of genes had at least one recombination event detected, and that 64.0% of genes had no detectable recombination. Could the author check that these number add up to 100%? $64\% + 38\% = 102\%$. The broad extent of homologous recombination detected in this H. influenzae population in consistent with previous estimates that place H. influenzae among the most recombining bacterial organisms, see Figure 2 in <https://pmc.ncbi.nlm.nih.gov/articles/PMC11067023/>. Could the authors provide an estimate of homologous recombination rate (r/m) for H. influenzae in their dataset?
- The authors should provide a summary of the geographical distribution of H. influenzae isolates in the global dataset, ideally on a world map, that highlights the biases in distribution inherent to an opportunistic dataset that most likely does not reflect the real regional prevalence and global distribution of H. influenzae strains.
- The following statement is hard to interpret: “595 clusters contained fewer than 10 isolates, leading to many larger clusters being paraphyletic according to the core genome tree, with the monophyletic clade containing small or singleton clusters, likely reflecting a change in the accessory genome of the smaller cluster which brought the pairwise distances above PopPUNK’s clustering threshold.”
- The following statement seems contradictory: on the one hand authors state that “Nucleotide diversity in both core ($n=1103$) and accessory ($n=8843$) genes was generally similar” to then state the opposite “although core genes are significantly less diverse than accessory genes”, please check and clarify this point.
- The authors claim that “a single universal vaccine could possibly be developed to combat invasive H. influenzae disease” based on the observation that H. influenzae core genes display rather low genetic diversity. An ideal vaccine antigen should indeed be conserved among all strains but also abundant epitopes on the bacterial surface, be immunogenic, and induce protective immune responses, as discussed here: <https://journals.asm.org/doi/full/10.1128/cvi.00089-15>. The authors may want to comment on the status of development of vaccines for non-typeable H. influenzae strains and, for any potential vaccine candidates, if these are part of the core-genome.

Data availability:

- The authors should provide a supplementary table that includes all isolates considered in this study, those sequenced in this study (PRJEB41043) and those in the global public collection identified, with the genomic QC metrics derived and used to filter

isolates, not only the isolates that passed QC currently present in the microreact project.

Methods:

- The section "Research involving human participants, their data, or biological material" of the Reporting Summary needs to be completed.

Minor changes:

- In the Abstract, in statement "Despite no Hib immunisation, serotype b was uncommonly found, while 91.7% of isolates were NT.", please indicate the exact percentage of serotype b cases found in this setting. When referring to NT in the Abstract, authors may want to refer to these cases as "unencapsulated non-typeable".

- In statement "Infants in the Maela cohort (Fig. 1) were predominantly colonised by NT H. influenzae, despite lacking immunisation against Hib.", please add the numbers supporting this statement, that is, total number of infants screened, and proportion of infants colonized by NT Hi.

- In statement "In silico capsule typing was generally congruent with the agglutination-based phenotypic serotyping". Can the authors provide an overall accuracy of the concordance of in silico serotyping (i.e., that detected by Hicap v.1.0.3) compare against phenotypic serotyping for Hi? Using data derived from their Maela dataset but also indicate the concordance reported by previous studies.

- Spell out the full name of the NPS abbreviation before using it. Should NPSS-STGG in statement "NPSS-STGG specimens were transferred to the SMRU" be NPS-STGG instead?

- In Figure 6, each phylogenetic tree could be labelled with their cluster number on the figure to facilitate identification.

- The following two Discussion statements seem to contradict each other: "NT H. influenzae are equally capable of causing invasive disease irrespective of their genetic background, even in a pre-Hib vaccine host population." and "While only colonising isolates were available from the Maela cohort". Both colonising and clinical pneumoniae isolates were included in this study, please ensure this point is clear.

- Delete repeated 'genomic' term in statement: "Widespread genomic genomic surveillance in such settings".

- In Discussion statement: "However, it is probably more likely that the difference in sampling strategy in the Maela data leads to higher statistical power to identify adaptation when it has occurred", do the authors mean that the higher density of sampling in the Maela camp led to higher statistical power to identify adaptation via dN/dS? Please clarify this point.

Decision Letter:

28th February 2025

Dear Jukka,

Thank you for your patience while your manuscript "The major pathogen Haemophilus influenzae experiences pervasive recombination and purifying selection at local and global scales" was under peer-review at Nature Microbiology. It has now been seen by 2 referees, whose expertise and comments you will find at the end of this email. Although they find your work of some potential interest, they have raised a number of concerns that will need to be addressed before we can consider publication of the work in Nature Microbiology.

In particular, both referee say that a clear definition of what the study wanted to achieve and what the key take-home messages are is currently missing. Further, referee #1 states that the longitudinally collected samples of H. influenzae and the publicly available WGS data should be analysed separately. This referee also suggests to make the distinction between serotype and non-typeable H. influenzae wherever this applies. Also, referee #1 says that some statements may need to be qualified when quantitative terms are applied. Referee #2 says that figures and proportions of H. influenzae isolates in the host deduplicated dataset should be provided. Further, the referee says that some statements are misleading. Referee #2 also says that it should be "attempted to build a core-genome phylogeny using a robust ML-based phylogenetics tool such as IQTree, using the genetic variation in the genes of the recently developed core genome multilocus sequence typing scheme for H. influenzae." Editorially, we will require all referee comments to be addressed in full.

Should further experimental data and text changes allow you to address these criticisms, we would be happy to look at a revised manuscript.

We strongly support public availability of data. Please place the data used in your paper into a public data repository, if one exists, or alternatively, present the data as Source Data or Supplementary Information. If data can only be shared on request, please explain why in your Data Availability Statement, and also in the correspondence with your editor. For some data types,

deposition in a public repository is mandatory - more information on our data deposition policies and available repositories can be found at <https://www.nature.com/nature-research/editorial-policies/reporting-standards#availability-of-data>.

Please include a data availability statement as a separate section after Methods but before references, under the heading "Data Availability". This section should inform readers about the availability of the data used to support the conclusions of your study. This information includes accession codes to public repositories (data banks for protein, DNA or RNA sequences, microarray, proteomics data etc...), references to source data published alongside the paper, unique identifiers such as URLs to data repository entries, or data set DOIs, and any other statement about data availability. At a minimum, you should include the following statement: "The data that support the findings of this study are available from the corresponding author upon request", mentioning any restrictions on availability. If DOIs are provided, we also strongly encourage including these in the Reference list (authors, title, publisher (repository name), identifier, year). For more guidance on how to write this section please see: <http://www.nature.com/authors/policies/data/data-availability-statements-data-citations.pdf>

* If you have not done so already we suggest that you begin to revise your manuscript so that it conforms to our Article format instructions at <http://www.nature.com/nmicrobiol/info/final-submission>. Refer also to any guidelines provided in this letter.

When submitting the revised version of your manuscript, please pay close attention to our [href="https://www.nature.com/nature-portfolio/editorial-policies/image-integrity">Digital Image Integrity Guidelines.](https://www.nature.com/nature-portfolio/editorial-policies/image-integrity) and to the following points below:

EXTENDED DATA FIGURES

Link Redacted

Note: This url links to your confidential homepage and associated information about manuscripts you may have submitted or be reviewing for us. If you wish to forward this e-mail to co-authors, please delete this link to your homepage first.

Nature Microbiology is committed to improving transparency in authorship. As part of our efforts in this direction, we are now requesting that all authors identified as 'corresponding author' on published papers create and link their Open Researcher and Contributor Identifier (ORCID) with their account on the Manuscript Tracking System (MTS), prior to acceptance. This applies to primary research papers only. ORCID helps the scientific community achieve unambiguous attribution of all scholarly contributions. You can create and link your ORCID from the home page of the MTS by clicking on 'Modify my Springer Nature account'. For more information please visit [please visit www.springernature.com/orcid](http://www.springernature.com/orcid).

If you wish to submit a suitably revised manuscript we would hope to receive it within 6 months. If you cannot send it within this time, please let us know.

Yours sincerely,

Reviewer Expertise:

Referee #1: Haemophilus-influenzae, infection in children, epidemiology
Referee #2: Epigenomics

Reviewer Comments:

Reviewer #1 (Remarks to the Author):

General comments

While this study provides useful information using whole genome sequence data from isolates of Haemophilus influenzae collected from the Maela camp for displaced persons in Northwestern Thailand and similar WGS data publicly available, there are parts of the manuscript that can be improved for clarity. Although it has been stated in the "Introduction" (lines 82-84) "As a consequence, the genetic population structure and evolutionary dynamics of the species remain poorly understood, particularly at a global scale", the objectives or aims of this study has not been clearly spelled out. Have the authors used their data to answer these questions, (a) the genetic population structure and (b) evolutionary dynamics of H. influenzae?

Also since the longitudinally collected samples of H. influenzae is likely to be different from H. influenzae population with publicly available WGS data (time of collection of isolates, geographical origin of the isolates, and social economic background of the host that yielded these isolates), the WGS data from these two apparently very different populations should be analyzed separately and their results compared rather than to look at as one population. By lumping the analysis together may potentially blur any findings of significance if it involves small number of isolates especially for isolates collected from the later host population.

Another important but not considered carefully by the authors is the fact that the genomic epidemiology of serotype H. influenzae is very different from that of non-typeable H. influenzae. Therefore, general statement like this "H. influenzae appears to have a population structure reminiscent of panmixia, where routine gene flow between members of the species prevents the formation of stable lineages" should be qualified by whether this applies to serotypeable or non-typeable H. influenzae as the former is monophyletic and the later is paraphyletic. I would suggest the authors to go over the whole manuscript and to make the distinction between serotype and non-typeable H. influenzae wherever this applies.

Some statements may need to be qualified when quantitative terms are applied. For example, this statement "Technically, clustering methods have likely been limited by the low levels of nucleotide diversity observed within the H. influenzae genome, particularly the core genome, even at a global scale" when referring to "low levels of nucleotide diversity, particularly the core genome" is somewhat puzzling since overall, the amount of genetic diversity is large as reflected in the number of unique MLST allelic profiles found in the Maelae samples (2344 isolates with allelic profiles not found in the MLST database and it is likely many of them are (a) non-typeable strains and (b) not of the same profile, although the exact serotype nature of the isolates yielding these new allelic profile and the exact number of unique allelic profile were not described by the authors. This is also evident in the large number of allelic sequences found in the H. influenzae MLST database (412,757 allele sequences from 10,999 isolates when the database was accessed on Dec 29, 2024).

Also, the description of a highly recombinant species (lines 166-167) and the low level of nucleotide diversity (line 286) needs to be addressed/discussed with a more comprehensive view.

Various terms were used to address the genetic or population structure of H. influenzae, such as clade (line 105), lineages (line 125) and clusters (line 128). Are these terms used consistently? Or defined in the manuscript.

Finally, it is unclear what is the conclusion of this study and does the conclusion address the objectives of this study.

Specific comments

- Line 38, in abstract, >4000, is it possible to provide the exact number here?
- Line 40-42, the pan-resistant lineages, were they NT or of all the Hi regardless of serotype?
- Table 1, was the data on serotype distribution based on in-silico from WGS data or agglutination based phenotype serotyping? Please add this information to the title of the Table.
- Line 106-110: accuracy of the in-silico capsule typing for serotype d and serotype e may not reflect the actual serotype nature of the isolates involved, there were three isolates identified by serotyping which means an error of either $3/8 = 37.5\%$ or $3/11 = 27.3\%$ for serotype d and for serotype e, the error maybe either $5/36 = 13.9\%$ or $5/41 = 12.3\%$. Therefore, the statement on lines 106-107, "In silico capsule typing was generally congruent with the agglutination-based phenotypic serotyping" should be qualified by adding the following, "except serotype d and serotype e".
- Line 106, "Two isolates gave only partial in silico capsule typing results", what were the serotyping results of these two isolates (which were included in Figure 2). Why is this information on serotype by phenotypic agglutination test not added in here?
- Line 122, please provide % for 2344/3970 of isolates; this means 58.9% of the H. influenzae isolates from Maela were made up of new sequence types not previously described elsewhere.
- Page 128, what is meant by "optimal clustering"?
- Page 127-131, does it mean PopPUNK grouped the 3970 isolates into 122 clusters but only 13 clusters contained at least 100 isolates and the largest cluster contained 349 isolates while 50% of the clusters (which means half of 122 clusters or 61 clusters) contained 10 or fewer isolates?
- Lines 129-131, the statement "with only 13 PopPUNK clusters in total each representing at least 100 isolates (out of a total of 122 clusters)", should it be consistent to describe or add that there were 20 clusters with >50 isolates? (as described in the legend for Figure 2)
- Lines 124-127: the data presented in Supplementary Figure 1 would be more informative and easier to read if presented as a table involving the columns of Clonal complex, Sequence Type, Serotype, and number of isolates per category. Right now it is

difficult to tell if the paraphyletic lineages involved serotypeable isolates or non-typeable isolates. Or keep Supplementary Figure 1 but also added the data as a supplementary Table.

- Line 138-141. It appears more isolates of Hib carried AMR determinants than NT-Hi (please describe % of Hib with AMR determinants versus % of NT-Hi with AMR determinants; also % of invasive Hib with AMR determinants versus % of carriage Hib with AMR determinants; % of invasive NT-Hi with AMR determinants versus % of carriage NT-Hi with AMR determinants; % of invasive H. influenzae with AMR determinants versus % of carriage H. influenzae with AMR determinants; an even better is to present this data by serotype (a, b, e, f, and non-typeable).
- Line 179-185: It is unclear if the largest cluster (composing of 483 isolates) was non-typeable or serotypeable? And the serotype nature of the smaller cluster (with 10 or fewer isolates) was also not described. These should be added here for completeness.
- Lines 180-181, regarding this statement "largest cluster composed of 483 isolates, and 19 clusters containing at least 100 isolates", what serotype is the largest cluster? And is it monophyletic or paraphyletic?
- Figure 3, the in-silico serotype would be easily to read if the were labelled by a letter instead of colour (since the colour legend was very difficult to read) especially if serotypes were usually restricted by their clustering within the phylogeny.
- Lines 196-197, is this statement correct? "extremely low level of nucleotide diversity evident during the recombination analysis of the Maela cohort", in view of this earlier statement, lines 166-167, "H. influenzae population within Maela is extremely Recombinant"?
- Lines 315-316, please refer to the data presented in the "Results Section" to support this statement "all but one of the pan-resistant MDR lineages was predominantly identified in the newly sequenced Maela isolates," (is this figure 6 and would need some elaboration).

Reviewer #2 (Remarks to the Author):

In this study, MacAlasdair and co-authors sequenced and analysed over 4,000 isolates of H. influenzae from an unvaccinated paediatric carriage and pneumonia cohort. This is the first time that a large collection of both carriage and disease isolates of the same population have been sequenced in a defined geographical region. Among the main findings, the authors found that serotype b was rarely found, with 91.7% of isolates being unencapsulated. They also found that the ability to cause invasive disease is not restricted to any subpopulation.

Major comments:

- An important observation of this study is that "non typeable isolates made up 91.7% of all isolates, and serotype b isolates are the second most prevalent, making up 5.7% of the population.". How generalisable is this observation? Can the authors contextualise this observation in the Discussion? are these proportions of NTHi and Hib similar in carriage and infection Hi of other strain populations?
- It seems numbers and proportions of H. influenzae isolates reported in the manuscript are indicated for the overall number of H. influenzae isolates that passed QC (denominator) but considering the dataset includes multiple longitudinal isolates per infant ("isolates were retrieved from a mother-infant cohort of 999 pregnant women from the Maela") the authors should also provide figures and proportions of H. influenzae isolates in the host deduplicated dataset.
- In Results section "Genetic population structure of H. influenzae in Maela": the authors conclude that "MLST sequence types (STs) and clonal complexes (CCs) [...] were unable to provide meaningful insight into the population structure." because the "majority of the Maela isolates (2,344/3,970) contained combinations of alleles not present in the current MLST database". This statement is misleading, the authors also state that STs and CCs do split the population into groups of closely related isolates, therefore the STs and CCs indeed provide some information on population structure. The lack of previously reported MLST profiles in the study population does not necessarily compromise the applicability of MLST typing more broadly. The identification of paraphyletic lineages of STs and CCs, interspersed with isolates from other STs/CCs, is instead what limits the applicability of MLST in H. influenzae. Please rephrase these statements to make this point clearer. See my next comment on the use of cgMLST.
- The authors should attempt to build a core-genome phylogeny using a robust ML-based phylogenetics tool such as IQTree, using the genetic variation in the genes of the recently developed core genome multilocus sequence typing scheme for H. influenzae: <https://www.microbiologyresearch.org/content/journal/mgen/10.1099/mgen.0.001281>
Can the authors observe a better correlation between PopPUNK clusters and core genome clusters?
- The Results section 'Distribution of AMR determinants' could be extended to report on the prevalence and distribution of acquired AMR mutations and genes in the phylogeny. Also, as far as I am aware, AMRFinderPlus does not support the detection of AMR specifically for H. influenzae, which may miss important AMR mutations. This limitation should be mentioned and discussed. The authors may want to consider the recent work of Matthias Merker 's group in detecting AMR from Haemophilus influenzae genomes (e.g., <https://pubmed.ncbi.nlm.nih.gov/35139905/> and <https://pubmed.ncbi.nlm.nih.gov/39633433/>). Also, in the microreact project metadata, include the specific AMR genetic determinants identified by AMRFinderPlus, not just if isolates have ('Yes') or not ('No') such determinants.
- In the Results section 'Quantification of homologous recombination' the authors quantified the extended of homologous recombination per gene across all genes in the pan-genome of H. influenzae. The authors found that 38% of genes had at least one recombination event detected, and that 64.0% of genes had no detectable recombination. Could the author check that these number add up to 100%? $64\% + 38\% = 102\%$. The broad extent of homologous recombination detected in this H. influenzae

population is consistent with previous estimates that place *H. influenzae* among the most recombining bacterial organisms, see Figure 2 in <https://pmc.ncbi.nlm.nih.gov/articles/PMC11067023/>. Could the authors provide an estimate of homologous recombination rate (r/m) for *H. influenzae* in their dataset?

- The authors should provide a summary of the geographical distribution of *H. influenzae* isolates in the global dataset, ideally on a world map, that highlights the biases in distribution inherent to an opportunistic dataset that most likely does not reflect the real regional prevalence and global distribution of *H. influenzae* strains.

- The following statement is hard to interpret: "595 clusters contained fewer than 10 isolates, leading to many larger clusters being paraphyletic according to the core genome tree, with the monophyletic clade containing small or singleton clusters, likely reflecting a change in the accessory genome of the smaller cluster which brought the pairwise distances above PopPUNK's clustering threshold."

- The following statement seems contradictory: on the one hand authors state that "Nucleotide diversity in both core ($n=1103$) and accessory ($n=8843$) genes was generally similar" to then state the opposite "although core genes are significantly less diverse than accessory genes", please check and clarify this point.

- The authors claim that "a single universal vaccine could possibly be developed to combat invasive *H. influenzae* disease" based on the observation that *H. influenzae* core genes display rather low genetic diversity. An ideal vaccine antigen should indeed be conserved among all strains but also abundant epitopes on the bacterial surface, be immunogenic, and induce protective immune responses, as discussed here: <https://journals.asm.org/doi/full/10.1128/cvi.00089-15>. The authors may want to comment on the status of development of vaccines for non-typeable *H. influenzae* strains and, for any potential vaccine candidates, if these are part of the core-genome.

Data availability:

- The authors should provide a supplementary table that includes all isolates considered in this study, those sequenced in this study (PRJEB41043) and those in the global public collection identified, with the genomic QC metrics derived and used to filter isolates, not only the isolates that passed QC currently present in the microreact project.

Methods:

- The section "Research involving human participants, their data, or biological material" of the Reporting Summary needs to be completed.

Minor changes:

- In the Abstract, in statement "Despite no Hib immunisation, serotype b was uncommonly found, while 91.7% of isolates were NT.", please indicate the exact percentage of serotype b cases found in this setting. When referring to NT in the Abstract, authors may want to refer to these cases as "unencapsulated non-typeable".

- In statement "Infants in the Maela cohort (Fig. 1) were predominantly colonised by NT *H. influenzae*, despite lacking immunisation against Hib.", please add the numbers supporting this statement, that is, total number of infants screened, and proportion of infants colonized by NT Hi.

- In statement "In silico capsule typing was generally congruent with the agglutination-based phenotypic serotyping". Can the authors provide an overall accuracy of the concordance of in silico serotyping (i.e., that detected by Hicap v.1.0.3) compared against phenotypic serotyping for Hi? Using data derived from their Maela dataset but also indicate the concordance reported by previous studies.

- Spell out the full name of the NPS abbreviation before using it. Should NPSS-STGG in statement "NPSS-STGG specimens were transferred to the SMRU" be NPS-STGG instead?

- In Figure 6, each phylogenetic tree could be labelled with their cluster number on the figure to facilitate identification.

- The following two Discussion statements seem to contradict each other: "NT *H. influenzae* are equally capable of causing invasive disease irrespective of their genetic background, even in a pre-Hib vaccine host population." and "While only colonising isolates were available from the Maela cohort". Both colonising and clinical pneumoniae isolates were included in this study, please ensure this point is clear.

- Delete repeated 'genomic' term in statement: "Widespread genomic genomic surveillance in such settings".

- In Discussion statement: "However, it is probably more likely that the difference in sampling strategy in the Maela data leads to higher statistical power to identify adaptation when it has occurred", do the authors mean that the higher density of sampling in the Maela camp led to higher statistical power to identify adaptation via dN/dS ? Please clarify this point.

Version 2:

Reviewer comments:

Reviewer #2

(Remarks to the Author)

General comments

The authors have made significant improvements to their manuscript.

To increase clarity and to put the subject matter in the perspective of disease potential and antibiotic resistance, I have a few comments below for the authors' consideration.

Specific comments

In the Introduction Section, it may be useful to write a few sentences pertaining to the importance of antibiotic resistance in *H. influenzae* and the emergence of multi-drug resistant strains, in order for readers to understand why multi-drug resistance is a concern for *H. influenzae*, especially when all sections in the manuscript from Materials & Methods to Results and Discussion all made reference to antibiotic resistance.

In the Results Section,

Under "Serotype distribution across the Maela paediatric population", it would be useful to include what number / % of samples (total 3970 before de-duplicated and 3210 for de-duplicated) were involved in pneumonia cases and what number / % of samples were from carriage? Similarly, the serotype distribution of pneumonia cases (or in other words of the samples from pneumonia cases, how many (%) were Hib, how many (%) were NT etc) and serotype distribution in samples from carriage?

Under "Genetic population structure of *H. influenzae* in Maela" (lines 120-135), it would be helpful if the authors can actually describe how many PopPUNK clusters were found in the overall Maela samples (of 3970 samples and 3210 de-duplicated samples) and how many belonged to serotypeable (may be helpful to separate serotype b from the other serotypes since this serotype accounted for 5.7% of samples) and how many PopPUNK clusters were found in the non-typeable samples? This separation of data would help readers to differentiate the results obtained from serotype versus non-typeable samples. Although the data may be found in the figures, it is easier to read a short summary text than trying to turn to the figures (the different colours used to denote the serotypes may not always be easier to tell; e.g. see figure 3).

Under "Distribution of AMR determinants" (lines 139-152), this statement regarding serotype b "In the Maela host-deduplicated dataset, most of the MDR isolates (resistance against at least four out of nine antibiotic classes, Methods) (507/3210), were NT (77.3%, 392/507), followed by serotype b (22.3%, 113/507), and two serotype e isolates. Hence, serotype b was clearly overrepresented among the more resistant isolates (overall frequency 4.8%, Table 1), while there were less NT (overall frequency 92.7%, Table 1)", this clearly shows the significance of Hib as this serotype appears to harbor more multi-drug resistance (113 MDR Hib out of a total of 154 Hib (Table 1) which makes 73.4% of Hib as being MDR. Personally I have not seen this level of MDR in any infectious bacterial pathogens. A simple phrase of Hib being over-represented (lines 146-147) appears inadequate here. I would like to see the authors clearly state what % of overall samples (n = 3120) being MDR (507/3210 = 15.8%) with 392 or 77.3% being NT, 113 or 22.3% being Hib and 2 or 0.4% being serotype e. Then stated what % of Hib were MDR (113/154 or 73.4%), what % of NT were MDR (392/2977 or 13.2%) and what % of serotype e being MDR (2/26 or 7.7%).

In lines 149-152, the authors stated that "Within pneumonia cases (523/3210), 17.0% (89/523) of isolates were MDR, of which 76 were NT, 12 of serotype b and one was serotype e; it would be helpful to include details for readers to understand the serotype distribution amongst the pneumonia cases? Or in other words, of the 523 pneumonia cases, what number and % were due to Hib, what number and % were due to NT, etc. Otherwise, the data is not completely disclosed for readers to have the full picture. Similarly, the same should be done for the "non-pneumonia cases" or should this be carriage cases? As how this was used elsewhere in the manuscript including in Figure 1?

Under "Quantification of homologous recombination", and line 165, suggest to consider rewording to "Of the 7015 genes in the pangenome of our samples"

In the Discussion Section,

The high rate of MDR in Hib is a concern and deserves some discussion on how meningitis caused by such MDR Hib can be problematic and how conjugate vaccine against Hib would be a useful tool to control MDR Hib. This then can lead into discussions on vaccination against NT-Hi.

Reviewer #3

(Remarks to the Author)

The authors have addressed all major comments. Specifically, they generated a core-genome ML phylogeny, which is mostly consistent with PopPunk clusters; they provide a map with the geographical distribution of isolates in the global dataset; they report the accuracy on in silico serotyping; and have improved the contextualisation of their findings. They have also created a supplementary table with the genome accessions and isolate metadata. All minor comments have also been addressed. I have no further comments or suggested changes.

Decision Letter:

Our ref: NMICROBIOL-24103124B

2nd September 2025

Dear Dr. Corander,

Thank you for submitting your revised manuscript "The major pathogen *Haemophilus influenzae* experiences pervasive recombination and purifying selection at local and global scales" (NMICROBIOL-24103124B). It has now been seen by the original referees and their comments are below. The reviewers find that the paper has improved in revision, and therefore we'll be happy in principle to publish it in *Nature Microbiology*, pending minor revisions to satisfy the referees' final requests and to comply with our editorial and formatting guidelines.

Thank you again for your interest in *Nature Microbiology*. Please do not hesitate to contact me if you have any questions.

Sincerely,

Reviewer #2 (Remarks to the Author):

General comments

The authors have made significant improvements to their manuscript.

To increase clarity and to put the subject matter in the perspective of disease potential and antibiotic resistance, I have a few comments below for the authors' consideration.

Specific comments

In the Introduction Section, it may be useful to write a few sentences pertaining to the importance of antibiotic resistance in *H. influenzae* and the emergence of multi-drug resistant strains, in order for readers to understand why multi-drug resistance is a concern for *H. influenzae*, especially when all sections in the manuscript from Materials & Methods to Results and Discussion all made reference to antibiotic resistance.

In the Results Section,

Under "Serotype distribution across the Maela paediatric population", it would be useful to include what number / % of samples (total 3970 before de-duplicated and 3210 for de-duplicated) were involved in pneumonia cases and what number / % of samples were from carriage? Similarly, the serotype distribution of pneumonia cases (or in other words of the samples from pneumonia cases, how many (%) were Hib, how many (%) were NT etc) and serotype distribution in samples from carriage?

Under "Genetic population structure of *H. influenzae* in Maela" (lines 120-135), it would be helpful if the authors can actually describe how many PopPUNK clusters were found in the overall Maela samples (of 3970 samples and 3210 de-duplicated samples) and how many belonged to serotypeable (may be helpful to separate serotype b from the other serotypes since this serotype accounted for 5.7% of samples) and how many PopPUNK clusters were found in the non-typeable samples? This separation of data would help readers to differentiate the results obtained from serotype versus non-typeable samples. Although the data may be found in the figures, it is easier to read a short summary text than trying to turn to the figures (the different colours used to denote the serotypes may not always be easier to tell; e.g. see figure 3).

Under "Distribution of AMR determinants" (lines 139-152), this statement regarding serotype b "In the Maela host-deduplicated dataset, most of the MDR isolates (resistance against at least four out of nine antibiotic classes, Methods) (507/3210), were NT (77.3%, 392/507), followed by serotype b (22.3%, 113/507), and two serotype e isolates. Hence, serotype b was clearly overrepresented among the more resistant isolates (overall frequency 4.8%, Table 1), while there were less NT (overall frequency 92.7%, Table 1)", this clearly shows the significance of Hib as this serotype appears to harbor more multi-drug resistance (113 MDR Hib out of a total of 154 Hib (Table 1) which makes 73.4% of Hib as being MDR. Personally I have not seen this level of MDR in any infectious bacterial pathogens. A simple phrase of Hib being over-represented (lines 146-147) appears inadequate here. I would like to see the authors clearly state what % of overall samples (n = 3120) being MDR (507/3210 = 15.8%) with 392 or 77.3% being NT, 113 or 22.3% being Hib and 2 or 0.4% being serotype e. Then stated what % of Hib were MDR (113/154 or 73.4%), what % of NT were MDR (392/2977 or 13.2%) and what % of serotype e being MDR (2/26 or 7.7%).

In lines 149-152, the authors stated that "Within pneumonia cases (523/3210), 17.0% (89/523) of isolates were MDR, of which 76 were NT, 12 of serotype b and one was serotype e.; it would be helpful to include details for readers to understand the serotype distribution amongst the pneumonia cases? Or in other words, of the 523 pneumonia cases, what number and % were due to Hib, what number and % were due to NT, etc. Otherwise, the data is not completely disclosed for readers to have the full picture. Similarly, the same should be done for the "non-pneumonia cases" or should this be carriage cases? As how this was used elsewhere in the manuscript including in Figure 1?

Under “Quantification of homologous recombination”, and line 165, suggest to consider rewording to “Of the 7015 genes in the pangenome of our samples”

In the Discussion Section,

The high rate of MDR in Hib is a concern and deserves some discussion on how meningitis caused by such MDR Hib can be problematic and how conjugate vaccine against Hib would be a useful tool to control MDR Hib. This then can lead into discussions on vaccination against NT-Hi.

Reviewer #3 (Remarks to the Author):

The authors have addressed all major comments. Specifically, they generated a core-genome ML phylogeny, which is mostly consistent with PopPunk clusters; they provide a map with the geographical distribution of isolates in the global dataset; they report the accuracy on in silico serotyping; and have improved the contextualisation of their findings. They have also created a supplementary table with the genome accessions and isolate metadata. All minor comments have also been addressed. I have no further comments or suggested changes.

Version 3:

Decision Letter:

2nd October 2025

Dear Jukka,

I am pleased to accept your Article "Genetic population structure of Haemophilus influenzae at local and global scales" for publication in Nature Microbiology. Thank you for having chosen to submit your work to us and many congratulations.

Authors may need to take specific actions to achieve compliance with funder and institutional open access mandates. If your research is supported by a funder that requires immediate open access (e.g. according to [a Plan S principles](https://www.springernature.com/gp/open-science/plan-s-compliance) or the [NIH public access policy](https://www.springernature.com/gp/open-science/us-federal-agency-compliance)) then you should select the gold OA route, and we will direct you to the compliant route where possible. Because authors warrant under our subscription licensing terms that they haven't committed to licensing any version of their article under a licence inconsistent with the terms of our agreement – including the applicable embargo period – publication under the subscription model isn't suitable for authors whose funders require no embargo.

If you have any questions about our publishing options, costs, Open Access requirements, or our legal forms, please contact

ASJournals@springernature.com

Congratulations once again and I look forward to seeing the article published.

With kind regards,

P.S. Click on the following link if you would like to recommend Nature Microbiology to your librarian
<http://www.nature.com/subscriptions/recommend.html#forms>

** Visit the Springer Nature Editorial and Publishing website at http://editorial-jobs.springernature.com?utm_source=ejP_NMicro_email&utm_medium=ejP_NMicro_email&utm_campaign=ejP_NMicro for more information about our career opportunities. If you have any questions please click [here](mailto:editorial.publishing.jobs@springernature.com).

Open Access This Peer Review File is licensed under a Creative Commons Attribution 4.0 International License, which permits use, sharing, adaptation, distribution and reproduction in any medium or format, as long as you give appropriate credit to the original author(s) and the source, provide a link to the Creative Commons license, and indicate if changes were made. In cases where reviewers are anonymous, credit should be given to 'Anonymous Referee' and the source. The images or other third party material in this Peer Review File are included in the article's Creative Commons license, unless indicated otherwise in a credit line to the material. If material is not included in the article's Creative Commons license and your intended use is not permitted by statutory regulation or exceeds the permitted use, you will need to obtain permission directly

from the copyright holder.

Dear Editor,

Thank you for the opportunity to revise our work. We apologize that the revision took several months to complete, this was primarily due to a large-scale IT crash and subsequent data loss that happened in May at the Sanger Institute, which forced us to redo the phylogenetic analyses and rerun some foundational analyses due to data loss. We have now completed all the requested additional analyses and carefully revised the manuscript according to the reviewer and editorial feedback.

On behalf of my co-authors,

Jukka Corander

Firstly, we would like to thank the reviewers for detailed and constructive comments and suggestions which enabled us to significantly improve the paper.

Reviewer Expertise:

Referee #1: Haemophilus-influenzae, infection in children, epidemiology

Referee #2: Epigenomics

Reviewer Comments:

Reviewer #1 (Remarks to the Author):

General comments

While this study provides useful information using whole genome sequence data from isolates of Haemophilus influenzae collected from the Maela camp for displaced persons in Northwestern Thailand and similar WGS data publicly available, there are parts of the manuscript that can be improved for clarity. Although it has been stated in the "Introduction" (lines 82-84) "As a consequence, the genetic population structure and evolutionary dynamics of the species remain poorly understood, particularly at a global scale", the objectives or aims of this study has not been clearly spelled out. Have the authors used their data to answer these questions, (a) the genetic population structure and (b) evolutionary dynamics of H. influenzae?

We agree that the text did not make clear the main message of the manuscript, and have edited and expanded parts of the abstract, Introduction and the Discussion to make this clearer.

Also since the longitudinally collected samples of H. influenzae is likely to be different from H. influenzae population with publicly available WGS data (time of collection of isolates, geographical origin of the isolates, and social economic background of the host that yielded these isolates), the WGS data from these two apparently very different populations should be analyzed separately and their results compared rather than to look at as one population. By lumping the analysis together may potentially blur any findings of significance if it involves small number of isolates especially for isolates collected from the later host population.

This is a valid point, but there may have been some confusion as it has already been done in the manuscript where appropriate. The manuscript has been edited to make this more clear, but we provide a brief summary here for clarity: the low-level recombination and association with disease analyses were performed using only the newly-sequenced data from the Maela camp. A separate phylogeny and whole-genome clustering were also inferred for only the Maela data, but in order to understand the global population of *H. influenzae*,

clustering and phylogeny inference was also performed on the combined collections. This was done to maximise the diversity in geographical origin and host population of the sampled isolates, in order to gain insight into the global population structure of the species inasmuch as is possible.

Another important but not considered carefully by the authors is the fact that the genomic epidemiology of serotype H. influenzae is very different from that of non-typeable H. influenzae. Therefore, general statement like this “H. influenzae appears to have a population structure reminiscent of panmixia, where routine gene flow between members of the species prevents the formation of stable lineages” should be qualified by whether this applies to serotypeable or non-typeable H. influenzae as the former is monophyletic and the latter is paraphyletic. I would suggest the authors to go over the whole manuscript and to make the distinction between serotype and non-typeable H. influenzae wherever this applies.

Thank you for highlighting this, the manuscript has been edited to make sure that the distinction between encapsulated and non-typable *H. influenzae* is highlighted whenever relevant.

Some statements may need to be qualified when quantitative terms are applied. For example, this statement “Technically, clustering methods have likely been limited by the low levels of nucleotide diversity observed within the H. influenzae genome, particularly the core genome, even at a global scale” when referring to “low levels of nucleotide diversity, particularly the core genome” is somewhat puzzling since overall, the amount of genetic diversity is large as reflected in the number of unique MLST allelic profiles found in the Maelae samples (2344 isolates with allelic profiles not found in the MLST database and it is likely many of them are (a) non-typeable strains and (b) not of the same profile, although the exact serotype nature of the isolates yielding these new allelic profile and the exact number of unique allelic profile were not described by the authors. This is also evident in the large number of allelic sequences found in the H. influenzae MLST database (412,757 allele sequences from 10,999 isolates when the database was accessed on Dec 29, 2024).

The number of MLST allelic profiles and overall nucleotide diversity are not perfectly correlated, it is possible for a large number of very low-frequency mutations to generate a large number of allelic profiles, even though nucleotide diversity across the pangenome is on average very low. As part of this revision, we have added cgMLST data, and this is indeed what we see. Despite the very low average nucleotide diversity we have calculated across the pangenome, the number of allelic mismatches is quite high due to many low frequency mutations. This has now been clarified in the context of cgMLST results.

Also, the description of a highly recombinant species (lines 166-167) and the low level of nucleotide diversity (line 286) needs to be addressed/discussed with a more comprehensive view.

We have amended the text to make this description more comprehensive.

Various terms were used to address the genetic or population structure of H. influenzae, such as clade (line 105), lineages (line 125) and clusters (line 128). Are these terms used consistently? Or defined in the manuscript.

This is an excellent observation regarding language precision in the manuscript, we have edited it in order to define and then consistently use “lineage” as “monophyletic group of isolates”, or “cluster” to specifically refer to isolates grouped together from the output of PopPUNK or cgMLST clustering methods, which typically and hopefully, but not always, correspond with our use of the word “lineage”.

Finally, it is unclear what is the conclusion of this study and does the conclusion address the objectives of this study.

Thank you for pointing this out. We have substantially expanded the Discussion to more explicitly state conclusions of the study, in particular linking them to the stated objectives in the introduction.

Specific comments

- Line 38, in abstract, >4000, is it possible to provide the exact number here?

Yes, this is now provided.

- Line 40-42, the pan-resistant lineages, were they NT or of all the Hi regardless of serotype?

This has now been specified.

- Table 1, was the data on serotype distribution based on in-silico from WGS data or agglutination based phenotypic serotyping? Please add this information to the title of the Table.

This information has been added. Note that the table is now placed at the end of the manuscript in accordance with the formatting request from the Editor.

- Line 106-110: accuracy of the in-silico capsule typing for serotype d and serotype e may not reflect the actual serotype nature of the isolates involved, there were three isolates identified by serotyping which means an error of either $3/8 = 37.5\%$ or $3/11 = 27.3\%$ for serotype d and for serotype e, the error maybe either $5/36 = 13.9\%$ or $5/41 = 12.3\%$. Therefore, the statement on lines 106-107, "In silico capsule typing was generally congruent with the agglutination-based phenotypic serotyping" should be qualified by adding the following, "except serotype d and serotype e".

Thanks for pointing this out, we have now corrected the statement to avoid giving a misleading impression.

- Line 106, "Two isolates gave only partial in silico capsule typing results", what were the serotyping results of these two isolates (which were included in Figure 2). Why is this information on serotype by phenotypic agglutination test not added in here?

This information has now been added.

- Line 122, please provide % for 2344/3970 of isolates; this means 58.9% of the H. influenzae isolates from Maela were made up of new sequence types not previously described elsewhere.

In line with another reviewer's comments, we have replaced MLST with cgMLST in the manuscript

- Page 128, what is meant by "optimal clustering"?

We have edited text to clarify that this refers to the clustering at the end of PopPUNK's optimization process

- Page 127-131, does it mean PopPUNK grouped the 3970 isolates into 122 clusters but only 13 clusters contained at least 100 isolates and the largest cluster contained 349 isolates while 50% of the clusters (which means half of 122 clusters or 61 clusters) contained 10 or fewer isolates?

Yes, edited for clarity

- Lines 129-131, the statement "with only 13 PopPUNK clusters in total each representing at least 100 isolates (out of a total of 122 clusters)", should it be consistent to describe or add that there were 20 clusters with >50 isolates? (as described in the legend for Figure 2)

We have added the information from the Figure 2 caption to the text here.

• Lines 124-127: the data presented in Supplementary Figure 1 would be more informative and easier to read if presented as a table involving the columns of Clonal complex, Sequence Type, Serotype, and number of isolates per category. Right now it is difficult to tell if the paraphyletic lineages involved serotypeable isolates or non-typeable isolates. Or keep Supplementary Figure 1 but also added the data as a supplementary Table.

We have replaced Supplementary Figure 1 with a supplementary table that has the following columns: PopPUNK cluster, number of isolates Serotype, cgMLST, and Mismatch 100 cgMLST complex.

• Line 138-141. It appears more isolates of Hib carried AMR determinants than NT-Hi (please describe % of Hib with AMR determinants versus % of NT-Hi with AMR determinants; also % of invasive Hib with AMR determinants versus % of carriage Hib with AMR determinants; % of invasive NT-Hi with AMR determinants versus % of carriage NT-Hi with AMR determinants; % of invasive H. influenzae with AMR determinants versus % of carriage H. influenzae with AMR determinants; an even better is to present this data by serotype (a, b, e, f, and non-typeable).

This is correct and we have now added the requested details to the AMR section of the Results.

• Line 179-185: It is unclear if the largest cluster (composing of 483 isolates) was non-typeable or serotypeable? And the serotype nature of the smaller cluster (with 10 or fewer isolates) was also not described. These should be added here for completeness.

Summarized serotype information has been added in this section for large clusters and isolates belonging to minor clusters.

• Lines 180-181, regarding this statement “largest cluster composed of 483 isolates, and 19 clusters containing at least 100 isolates”, what serotype is the largest cluster? And is it monophyletic or paraphyletic?

Cluster 1, the largest PopPUNK cluster is predominantly serotype a, and is monophyletic. This information has been added to the text in this section.

• Figure 3, the in-silico serotype would be easily to read if the were labelled by a letter instead of colour (since the colour legend was very difficult to read) especially if serotypes were usually restricted by their clustering within the phylogeny.

While serotypes largely cluster together on the phylogeny, there is enough variation to make accurate direct text labels of the tip very difficult to read – the text labels are either highly overlapping or extremely small. To make the figure clearer, we have made the legend significantly larger, and chosen a more distinct colour palette and specifically with more contrast between nontypable and serotyped isolates.

• Lines 196-197, is this statement correct? “extremely low level of nucleotide diversity evident during the recombination analysis of the Maela cohort”, in view of this earlier statement, lines 166-167, “H. influenzae population within Maela is extremely Recombinant”?

Yes, this statement is correct. Our results from the Maela linkage disequilibrium analysis in particular, where the *H. influenzae* isolates had the fastest LD decay, or least LD between SNPs, shows that recombination levels are so high that recombination events are reducing the overall level of diversity in the population. Recombinant DNA is not introducing SNPs, it is more likely to be removing them.

• Lines 315-316, please refer to the data presented in the “Results Section” to support this statement “all but one of the pan-resistant MDR lineages was predominantly identified in the newly sequenced Maela isolates,” (is this figure 6 and would need some elaboration).

We have revised this statement and added further information to the Figure 6 caption regarding the origins of the isolates in pan-resistant lineages.

Reviewer #2 (Remarks to the Author):

In this study, MacAlasdair and co-authors sequenced and analysed over 4,000 isolates of *H. influenzae* from an unvaccinated paediatric carriage and pneumonia cohort. This is the first time that a large collection of both carriage and disease isolates of the same population have been sequenced in a defined geographical region. Among the main findings, the authors found that serotype b was rarely found, with 91.7% of isolates being unencapsulated. They also found that the ability to cause invasive disease is not restricted to any subpopulation.

Major comments:

- An important observation of this study is that “non typable isolates made up 91.7% of all isolates, and serotype b isolates are the second most prevalent, making up 5.7% of the population.”. How generalisable is this observation? Can the authors contextualise this observation in the Discussion? are these proportions of NTHi and Hib similar in carriage and infection Hi of other strain populations?

We thank you for bringing this up and have now included additional literature and amended the Discussion as suggested.

- It seems numbers and proportions of *H. influenzae* isolates reported in the manuscript are indicated for the overall number of *H. influenzae* isolates that passed QC (denominator) but considering the dataset includes multiple longitudinal isolates per infant (“isolates were retrieved from a mother-infant cohort of 999 pregnant women from the Maela”) the authors should also provide figures and proportions of *H. influenzae* isolates in the host deduplicated dataset.

We now provide these numbers also for the deduplicated data.

- In Results section “Genetic population structure of *H. influenzae* in Maela”: the authors conclude that “MLST sequence types (STs) and clonal complexes (CCs) [...] were unable to provide meaningful insight into the population structure.” because the “majority of the Maela isolates (2,344/3,970) contained combinations of alleles not present in the current MLST database”. This statement is misleading, the authors also state that STs and CCs do split the population into groups of closely related isolates, therefore the STs and CCs indeed provide some information on population structure. The lack of previously reported MLST profiles in the study population does not necessarily compromise the applicability of MLST typing more broadly. The identification of paraphyletic lineages of STs and CCs, interspersed with isolates from other STs/CCs, is instead what limits the applicability of MSLT in *H. influenzae*. Please rephrase these statements to make this point clearer. See my next comment on the use of cgMLST.

We have replaced the use of MLST and clonal complexes with cgMLST and network-based cgMLST mismatch clustering at 100 and 250 mismatches, in keeping with previously published work using cgMLST to study *H. influenzae*. The problematic statements in the manuscript have been removed or rewritten accordingly.

- The authors should attempt to build a core-genome phylogeny using a robust ML-based phylogenetics tool such as IQTree, using the genetic variation in the genes of the recently developed core genome multilocus sequence typing scheme for *H. influenzae*:

<https://www.microbiologyresearch.org/content/journal/mgen/10.1099/mgen.0.001281>

Can the authors observe a better correlation between PopPUNK clusters and core genome clusters?

Regarding the ML phylogenetic tree, this is a good suggestion which we previously had not done for the phylogeny of the entire combined dataset due to computation resource constraints. In the revised draft of the manuscript, we have now done this using IQ-TREE and a recombination-free Panaroo core genome alignment. We have also (largely) replaced the use of MLST in the paper with the use of cgMLST. This is discussed in more detail in the revised manuscript, but while cgMLST does perform better than traditional MLST, the high levels of recombination in *H. influenzae* still make the use of any kind of typing method unreliable on larger scales. Indeed we do not report the 500 allelic mismatch cgMLST clustering threshold as previous work has done, as that resulted in the entire dataset being composed of a single cluster.

- The Results section 'Distribution of AMR determinants' could be extended to report on the prevalence and distribution of acquired AMR mutations and genes in the phylogeny. Also, as far as I am aware, AMRFinderPlus does not support the detection of AMR specifically for *H. influenzae*, which may miss important AMR mutations. This limitation should be mentioned and discussed. The authors may want to consider the recent work of Matthias Merker 's group in detecting AMR from *Haemophilus influenzae* genomes (e.g., <https://pubmed.ncbi.nlm.nih.gov/35139905/> and <https://pubmed.ncbi.nlm.nih.gov/39633433/>). Also, in the microreact project metadata, include the specific AMR genetic determinants identified by AMRFinderPlus, not just if isolates have ('Yes') or not ('No') such determinants.

We thank the reviewers for the notification. Curated *H. influenzae*-specific AMR determinants have indeed been included in the AMRFinderPlus database as of v.2024-11-22.1. To address the limitation with the previous database, AMRFinderPlus was rerun using a more recent database version.

- In the Results section 'Quantification of homologous recombination' the authors quantified the extended of homologous recombination per gene across all genes in the pan-genome of *H. influenzae*. The authors found that 38% of genes had at least one recombination event detected, and that 64.0% of genes had no detectable recombination. Could the author check that these number add up to 100%? $64\% + 38\% = 102\%$. The broad extent of homologous recombination detected in this *H. influenzae* population is consistent with previous estimates that place *H. influenzae* among the most recombining bacterial organisms, see Figure 2 in <https://pmc.ncbi.nlm.nih.gov/articles/PMC11067023/>. Could the authors provide an estimate of homologous recombination rate (r/m) for *H. influenzae* in their dataset?

Our results are highly consistent with previous work indicating that *H. influenzae* has a very high recombination rate, however it is effectively impossible to accurately estimate r/m from our data. Part of this is the inherent difficulty in estimating r/m in any context, but particularly with the Maela camp data, which might otherwise seem like a good dataset in which to attempt to estimate r/m , our results indicate that the level of recombination is so high, that it is impossible to accurately estimate the mutation rate (m), due to recombination being frequent enough to remove mutations from our sampling. Such recombination is therefore completely undetectable, as it does not carry mutations from one genetic background into another, which can be detected when comparing enough samples. Recombination events between extremely similar sequences of DNA generally leave no traces to be detectable in the sequence data, and may actually inhibit the sampling of single nucleotide point mutations. As a result, any estimate of r/m we could generate from the data will be highly imprecise and likely inaccurate.

- The authors should provide a summary of the geographical distribution of *H. influenzae* isolates in the global dataset, ideally on a world map, that highlights the biases in distribution inherent to an opportunistic dataset that most likely does not reflect the real regional prevalence and global distribution of *H. influenzae* strains.

This is a great observation about a missing figure which was not obvious to us when writing the paper as we are so familiar with the data. We have now included a global map indicating the origins of the isolates, as well as emphasising the biases in the distribution in the text.

- The following statement is hard to interpret: "595 clusters contained fewer than 10 isolates, leading to many larger clusters being paraphyletic according to the core genome tree, with the monophyletic clade containing small or singleton clusters, likely reflecting a change in the accessory genome of the smaller cluster which brought the pairwise distances above PopPUNK's clustering threshold."

We have edited the text to make the interpretation clearer.

- The following statement seems contradictory: on the one hand authors state that "Nucleotide diversity in both core (n=1103) and accessory (n=8843) genes was generally similar" to then state the opposite "although core genes are significantly less diverse than accessory genes", please check and clarify this point.

Thank you for pointing out this poorly-worded sentence. It is meant to contrast range with central tendency, and has been rewritten as follows to do so: "Although nucleotide diversity in both core (n=1103) and accessory (n=8843) genes have overlapping ranges, core genes are on average significantly less diverse than accessory genes."

- The authors claim that "a single universal vaccine could possibly be developed to combat invasive H. influenzae disease" based on the observation that H. influenzae core genes display rather low genetic diversity. An ideal vaccine antigen should indeed be conserved among all strains but also abundant epitopes on the bacterial surface, be immunogenic, and induce protective immune responses, as discussed here: <https://journals.asm.org/doi/full/10.1128/cvi.00089-15>. The authors may want to comment on the status of development of vaccines for non-typeable H. influenzae strains and, for any potential vaccine candidates, if these are part of the core-genome.

Excellent suggestion, we have amended the discussion accordingly.

Data availability:

- The authors should provide a supplementary table that includes all isolates considered in this study, those sequenced in this study (PRJEB41043) and those in the global public collection identified, with the genomic QC metrics derived and used to filter isolates, not only the isolates that passed QC currently present in the microreact project.

This has been added.

Methods:

- The section "Research involving human participants, their data, or biological material" of the Reporting Summary needs to be completed.

We have completed the Reporting Summary.

Minor changes:

- In the Abstract, in statement "Despite no Hib immunisation, serotype b was uncommonly found, while 91.7% of isolates were NT.", please indicate the exact percentage of serotype b cases found in this setting. When referring to NT in the Abstract, authors may want to refer to these cases as "unencapsulated non-typeable".

Both additions have been made.

- In statement “Infants in the Maela cohort (Fig. 1) were predominantly colonised by NT H. influenzae, despite lacking immunisation against Hib.”, please add the numbers supporting this statement, that is, total number of infants screened, and proportion of infants colonized by NT Hi.

We have added citation to Table 1 to support the statement and also mention the host deduplication in this context.

- In statement “In silico capsule typing was generally congruent with the agglutination-based phenotypic serotyping”. Can the authors provide an overall accuracy of the concordance of in silico serotyping (i.e., that detected by Hicap v.1.0.3) compare against phenotypic serotyping for Hi?

In the light of the comment also from the other reviewer, we have now provided a more detailed description of the accuracy/concordance.

Using data derived from their Maela dataset but also indicate the concordance reported by previous studies.

We have added a comment on the level of concordance with previous studies.

- Spell out the full name of the NPS abbreviation before using it. Should NPSS-STGG in statement “NPSS-STGG specimens were transferred to the SMRU” be NPS-STGG instead?

Full name added and abbreviation corrected.

- In Figure 6, each phylogenetic tree could be labelled with their cluster number on the figure to facilitate identification.

These labels have been added to the figure.

- The following two Discussion statements seem to contradict each other: “NT H. influenzae are equally capable of causing invasive disease irrespective of their genetic background, even in a pre-Hib vaccine host population.” and “While only colonising isolates were available from the Maela cohort”. Both colonising and clinical pneumoniae isolates were included in this study, please ensure this point is clear.

Thank you for pointing out this ambiguity, we have carefully revised this part of Discussion to make sure the point is clear to the reader.

- Delete repeated ‘genomic’ term in statement: “Widespread genomic genomic surveillance in such settings”.

Corrected.

- In Discussion statement: “However, it is probably more likely that the difference in sampling strategy in the Maela data leads to higher statistical power to identify adaptation when it has occurred”, do the authors mean that the higher density of sampling in the Maela camp led to higher statistical power to identify adaptation via dN/dS? Please clarify this point.

Yes, and we have edited this for clarity.

Response to reviewer comments

Reviewer #2:

Remarks to the Author:

General comments

The authors have made significant improvements to their manuscript.

To increase clarity and to put the subject matter in the perspective of disease potential and antibiotic resistance, I have a few comments below for the authors' consideration.

Specific comments

In the Introduction Section, it may be useful to write a few sentences pertaining to the importance of antibiotic resistance in *H. influenzae* and the emergence of multi-drug resistant strains, in order for readers to understand why multi-drug resistance is a concern for *H. influenzae*, especially when all sections in the manuscript from Materials & Methods to Results and Discussion all made reference to antibiotic resistance.

This has been added as suggested.

In the Results Section,

Under "Serotype distribution across the Maela paediatric population", it would be useful to include what number / % of samples (total 3970 before de-duplicated and 3210 for de-duplicated) were involved in pneumonia cases and what number / % of samples were from carriage? Similarly, the serotype distribution of pneumonia cases (or in other words of the samples from pneumonia cases, how many (%) were Hib, how many (%) were NT etc) and serotype distribution in samples from carriage?

This has been addressed by amending the information presented in Table 1.

Under "Genetic population structure of *H. influenzae* in Maela" (lines 120-135), it would be helpful if the authors can actually describe how many PopPUNK clusters were found in the overall Maela samples (of 3970 samples and 3210 de-duplicated samples) and how many belonged to serotypeable (may be helpful to separate serotype b from the other serotypes since this serotype accounted for 5.7% of samples) and how many PopPUNK clusters were found in the non-typeable samples? This separation of data would help readers to differentiate the results obtained from serotype versus non-typeable samples. Although the data may be found in the figures, it is easier to read a short summary text than trying to turn to the figures (the different colours used to denote the serotypes may not always be easier to tell; e.g. see figure 3).

We have revised the text to make it more informative.

Under "Distribution of AMR determinants" (lines 139-152), this statement regarding serotype b "In the Maela host-deduplicated dataset, most of the MDR isolates (resistance against at least four out of nine antibiotic classes, Methods) (507/3210), were NT (77.3%, 392/507), followed by serotype b (22.3%, 113/507), and two serotype e isolates. Hence, serotype b was clearly overrepresented among the more resistant isolates (overall frequency 4.8%, Table 1), while there were less NT (overall frequency

92.7%, Table 1)”, this clearly shows the significance of Hib as this serotype appears to harbor more multi-drug resistance (113 MDR Hib out of a total of 154 Hib (Table 1) which makes 73.4% of Hib as being MDR. Personally I have not seen this level of MDR in any infectious bacterial pathogens. A simple phrase of Hib being over-represented (lines 146-147) appears inadequate here. I would like to see the authors clearly state what % of overall samples (n = 3120) being MDR (507/3210 = 15.8%) with 392 or 77.3% being NT, 113 or 22.3% being Hib and 2 or 0.4% being serotype e. Then stated what % of Hib were MDR (113/154 or 73.4%), what % of NT were MDR (392/2977 or 13.2%) and what % of serotype e being MDR (2/26 or 7.7%).

We have added the requested information as a supplementary table.

In lines 149-152, the authors stated that “Within pneumonia cases (523/3210), 17.0% (89/523) of isolates were MDR, of which 76 were NT, 12 of serotype b and one was serotype e; it would be helpful to include details for readers to understand the serotype distribution amongst the pneumonia cases? Or in other words, of the 523 pneumonia cases, what number and % were due to Hib, what number and % were due to NT, etc. Otherwise, the data is not completely disclosed for readers to have the full picture. Similarly, the same should be done for the “non-pneumonia cases” or should this be carriage cases? As how this was used elsewhere in the manuscript including in Figure 1?

We have clarified this matter in the text and by adding more information into Table 1.

Under “ Quantification of homologous recombination”, and line 165, suggest to consider rewording to “Of the 7015 genes in the pangenome of our samples”

This is now reworded.

In the Discussion Section,

The high rate of MDR in Hib is a concern and deserves some discussion on how meningitis caused by such MDR Hib can be problematic and how conjugate vaccine against Hib would be a useful tool to control MDR Hib. This then can lead into discussions on vaccination against NT-Hi.

It is already widely understood how the conjugate Hib vaccine is useful for controlling Hib meningitis and most countries have adopted it into their national programs. Given the length restriction imposed by the journal we have chosen to omit adding this.